# Bipolar filaments of human nonmuscle myosin 2-A and 2-B have distinct motile and mechanical properties

Luca Melli[1]*, Neil Billington[1], Sara A Sun[1], Jonathan E Bird[2†], Attila Nagy[3], Thomas B Friedman[2], Yasuharu Takagi[1], James R Sellers[1]*

[1]Cell Biology and Physiology Center, National Heart, Lung and Blood Institute, National Institutes of Health, Bethesda, United States; [2]Laboratory of Molecular Genetics, National Institute on Deafness and Other Communication Disorders, National Institutes of Health, Bethesda, United States; [3]Vaccine Production Program Laboratory, National Institutes of Allergy and Infectious Diseases, National Institutes of Health, Gaithersburg, United States

**Abstract** Nonmusclemyosin 2 (NM-2) powers cell motility and tissue morphogenesis by assembling into bipolar filaments that interact with actin. Although the enzymatic properties of purified NM-2 motor fragments have been determined, the emergent properties of filament ensembles are unknown. Using single myosin filament in vitro motility assays, we report fundamental differences in filaments formed of different NM-2 motors. Filaments consisting of NM2-B moved processively along actin, while under identical conditions, NM2-A filaments did not. By more closely mimicking the physiological milieu, either by increasing solution viscosity or by co-polymerization with NM2-B, NM2-A containing filaments moved processively. Our data demonstrate that both the kinetic and mechanical properties of these two myosins, in addition to the stochiometry of NM-2 subunits, can tune filament mechanical output. We propose altering NM-2 filament composition is a general cellular strategy for tailoring force production of filaments to specific functions, such as maintaining tension or remodeling actin.
DOI: https://doi.org/10.7554/eLife.32871.001

*For correspondence:
luca.melli@nih.gov (LM);
sellersj@nhlbi.nih.gov (JRS)

Present address: †Department of Pharmacology and Therapeutics, University of Florida College of Medicine, Gainesville, United States

Competing interests: The authors declare that no competing interests exist.

## Introduction

Humans have 13 class II myosin heavy chain genes. Most of these encode for the heavy chains of skeletal, cardiac and smooth muscle myosins, but three encode for the heavy chains of paralogs that are widely expressed in nonmuscle tissue. These are commonly called NM2-A, NM2-B and NM2-C, each with different heavy chains encoded by the *MYH9*, *MYH10* and *MYH14* genes, respectively (*Berg et al., 2001*; *Vicente-Manzanares et al., 2009*). Nonmuscle class II myosins (NM2) are molecular motors involved in cytokinesis, cell migration, adhesion and tissue morphogenesis (*Heissler and Manstein, 2013*; *Vicente-Manzanares et al., 2009*). The coiled-coil tail region of the heavy chain of each of these myosins homodimerizes and the neck region associates with an essential light chain (ELC) and a regulatory light chain (RLC), creating a hexameric molecule. These individual myosin molecules further self-associate via their tails to form bipolar filaments that are approximately 300 nm in length and contain, on average, either 30 myosin molecules for NM2-A and NM2-B or 16 myosin molecules for NM2-C (*Billington et al., 2013*; *Niederman and Pollard, 1975*). In addition, it has been demonstrated that NM2-A and NM2-B can co-polymerize to form heterotypic filaments in cells (*Beach et al., 2014*; *Shutova et al., 2014*). It is unknown why mammalian cells express three different NM2 paralogs, or what their individual or shared functions are (*Conti et al., 2008*; *Vicente-Manzanares et al., 2009*). The enzymatic activity and the filament assembly of each of the NM2

paralogs are regulated by phosphorylation of the RLC by myosin light chain kinase (MLCK), or other cellular kinases (*Heissler and Sellers, 2016*).

It has long been known that the different skeletal muscle myosins-2 have distinct enzymatic and mechanical properties from studies in muscle fibers and of isolated proteins (*Bottinelli and Reggiani, 2000*). Similarly, numerous enzymatic studies conducted on the soluble, single-headed subfragment-one (S1) or double-headed heavy meromyosin (HMM) fragments have revealed differences in the steady state and transient state kinetics of the three NM2 paralogs that suggests there is differentiation of function amongst them (*Heissler and Manstein, 2011*; *Kovács et al., 2003*; *Rosenfeld et al., 2003*; *Wang et al., 2003*). The enzymatic activity of all three NM2 paralogs is low compared to other myosin 2 family members. Nevertheless, of the three NM2 paralogs, NM2-A has the highest actin-activated ATPase activity and translocates actin filaments the fastest (*Kim et al., 2005*; *Wang et al., 2003*). During the course of binding and hydrolysis of ATP and subsequently product dissociation, myosin cycles through conformations that bind weakly and strongly to actin. The duty ratio (r) of a myosin is defined as the fraction of a kinetic cycle the myosin spends in a conformational state which binds strongly to actin. NM2-B has a four-fold higher duty ratio than NM2-A, meaning that it will spend a greater proportion of its ATPase cycle strongly bound to actin filaments (*Kovács et al., 2003*; *Wang et al., 2003*).

The role of myosin bipolar filaments as force generating structures in cells has been clearly demonstrated (*Vicente-Manzanares et al., 2009*). Myosin filaments can interact with actin filaments of opposite polarity to create a sarcomeric type contractile unit, similar to those found in striated muscles (*Ebrahim et al., 2013*). They may also move processively along actin filaments in cells which could aid in the organization of the actin cytoskeleton (*Beach et al., 2017*; *Ebrahim et al., 2013*; *Fenix et al., 2016*; *Hu et al., 2017*) or may serve as protein scaffolds (*Joo et al., 2007*; *Maddox et al., 2007*; *Piekny and Glotzer, 2008*). Recent studies using super-resolution fluorescence microscopy for the visualization of single, or small clusters of myosin filaments in cells, allow for the possibility of dissecting their detailed function (*Beach et al., 2017*; *Beach and Hammer, 2015*; *Beach et al., 2014*; *Fenix et al., 2016*; *Hu et al., 2017*). However, despite their fundamental roles in generating force in cells, the mechanical properties of individual NM2 filaments are unknown.

To understand their individual mechanical properties, we used purified full-length human NM2-A and NM2-B molecules expressed in *Sf9* cells to reconstitute and study the ability of single filaments of NM2 myosin paralogs to move along actin filaments bound to the coverslip surface. We find that NM2-B filaments are processive even at low-solution viscosity and that 5 – 10 myosin motor domains per half filament are required to ensure processivity. In contrast, NM2-A filaments are only processive when the viscosity of the solution is raised to mimic that of the intracellular millieu using methylcellulose or when NM2-A is co-polymerized with NM2-B. These data suggest that NM2-B is more adapted to a role of maintaining a high static tension level in cells, whereas NM2-A is geared to more rapid motile activity. Our data further show that formation of heterotypic NM2 filaments is an effective mechanism to fine tune filament mechanical properties between these two modes. As cell biologists start to decipher the cellular roles of NM2 filaments using fluorescence microscopy, our study demonstrates that not all NM2 filaments are equivalent and that subunit composition must be considered in order to fully understand their function.

## Results

### NM2-B filaments move processively on actin filaments

In a first set of experiments, we investigated the motility of NM2-B filaments using TIRF microscopy. We introduced a HaloTag at the N-terminus of the myosin heavy chain to allow for covalent labeling with dyes of different colors (*Figure 1—figure supplement 1*). The regulatory light chain (RLC) was phosphorylated using MLCK in order to activate the enzymatic activity of the myosin and to facilitate filament formation. We found that individual HaloTag-NM2-B filaments moved processively along fluorescently labeled actin filaments attached to a glass coverslip (*Figure 1A and B*; *Video 1*). The NM2-B filaments were slightly longer (~300 nm) than the diffraction limited resolution of our TIRF microscope and usually appeared as slightly elongated objects whose long and short axes could be readily distinguished. In the presence of 1 mM ATP, HaloTag-NM2-B filaments showed robust

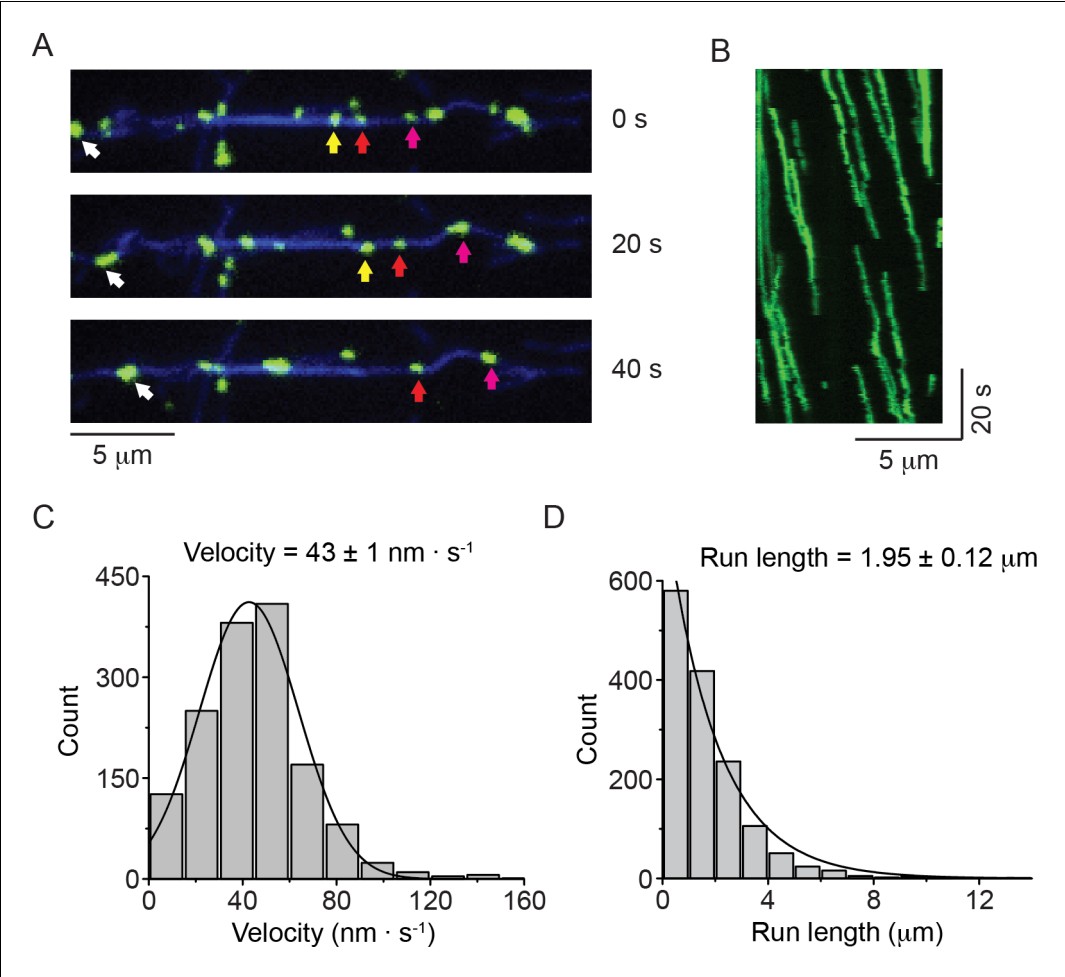

**Figure 1.** Nonmuscle myosin 2-B filaments move processively along actin filaments. (**A**) Movie frames showing HaloTag-NM2-B filaments (individual filaments marked by colored arrows), moving along Alexa Fluor 647 phalloidin labeled actin filaments (blue). (**B**) Kymograph for NM2-B filaments showing clear processive movement (diagonal lines) and long run lengths. (**C**) Frequency distribution histogram of NM2-B filament velocity. Black line is the Gaussian fit to the data yielding a velocity of $43 \pm 1$ nm $\cdot$ s$^{-1}$ (mean $\pm$ SEM; SD = 27 nm $\cdot$ s$^{-1}$, $R^2$ = 0.98). (**D**) Frequency distribution histogram of NM2-B filament run length. Black line is the single exponential fit to the data. The characteristic velocity and run length with their standard errors obtained from the fits are indicated in the graphs (n = 1463, $R^2$ = 0.99).

DOI: https://doi.org/10.7554/eLife.32871.002

The following source data and figure supplements are available for figure 1:

**Source data 1.** Data for the velocity (panel C) and run length (panel D).
DOI: https://doi.org/10.7554/eLife.32871.007

**Figure supplement 1.** GFP and HaloTags do not significantly interfere with myosin filament structure or motile function.
DOI: https://doi.org/10.7554/eLife.32871.003

**Figure supplement 1—source data 1.** Data for the velocity distributions shown in panel C.
DOI: https://doi.org/10.7554/eLife.32871.004

**Figure supplement 2.** GFP-RLC-NM2-B filaments move processively along actin filaments.
DOI: https://doi.org/10.7554/eLife.32871.005

**Figure supplement 2—source data 2.** Run length data for GFP-RLC NM2-B.
DOI: https://doi.org/10.7554/eLife.32871.006

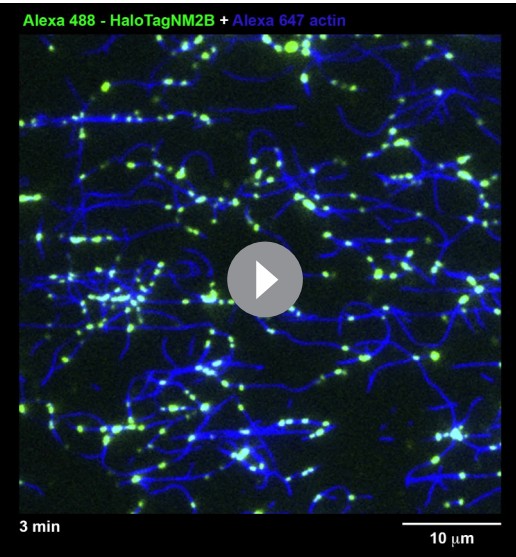

**Alexa 488 - HaloTagNM2B + Alexa 647 actin**

3 min                                          10 μm

**Video 1.** Movement of NM2-B filaments on surface bound actin filaments. The movie shows Alexa 488 labelled HaloTag-NM2-B filaments (green) moving along surface immobilized Alexa-647-labeled actin filaments (blue).

DOI: https://doi.org/10.7554/eLife.32871.008

processive motility with a characteristic run length of 1.95 ± 0.12 μm (mean ± SEM) and a velocity of 43 ± 21 nm⋅ s$^{-1}$ (mean ± SD, n = 1463 filaments) (*Figure 1C and D*). This experiment confirms an earlier study showing that NM2-B filaments purified with an N-terminally fused GFP-RLC (GFP-RLC-NM2-B) also moved processively along actin (*Nagy et al., 2013*). To directly compare the movements of myosin filaments using these two labeling strategies, we re-examined the movement of GFP-RLC-NM2-B filaments under the same conditions (*Figure 1—figure supplement 1*, *Figure 1—figure supplement 2*, *Video 2*). GFP-RLC-NM2-B filaments showed the same level of processivity as HaloTag-NM2-B filaments. Both types of myosin labeling are being used in cell biological studies (*Beach et al., 2017*; *Bruun et al., 2017*).

Several interesting features of the NM2-B filament motility were observed in our timelapse recordings, reflecting possible behaviors of NM2 filaments in cells. A single NM2-B filament could move on actin with its long axis perpendicular or parallel to the actin filament and could flip back and forth between these two modes while moving (*Video 2*). In the case of perpendicular interactions only one side of the myosin filament interacted with actin. The polarity of the actin filament determines the direction the bipolar myosin filament moves even when the NM2 filament is interacting in a parallel manner with the actin filament where motors from each end of the bipolar filament are in proximity to actin. When NM2-B filaments encountered an intersection of two actin filaments, they often paused before continuing along the original filament or switching to the other (*Video 3*). Since the actin filaments in this assay were tethered to the coverslip surface this would be analogous to the myosin exerting isometric tension on two actin filaments in a cell. In support of this, when a NM2-B filament was bound to a surface-tethered actin filament, we observed that it could simultaneously propel the sliding of a free actin filament (*Video 4*). This represents an example for how NM2 filaments can remodel actin cytoskeletal networks, such as those in lamellipodia.

## Merging and splitting of multi-filament structures

In cells, individual myosin filaments can align vertically to form highly registered stacks (*Fenix et al., 2016*; *Hu et al., 2017*; *Shutova et al., 2014*; *Verkhovsky et al., 1995*). Stacking of filaments was also occasionally observed in preparations of pure NM2-B in the electron microscope (*Billington et al., 2013*). *Video 5* shows an example where a bright stack of NM2-B myosin filaments landed next to an immobilized actin filament. An individual myosin filament unit delaminated from the stack and moved along the actin filament. Later in the same movie, another individual myosin filament landed on the actin filament, moves toward the stack and appeared to join it. We found that multi-filament structures dynamically formed in our in vitro motility assays when one myosin filament encountered another while moving along an actin filament (*Video 6*). In this movie, biotinylated-actin filaments bound to the surface were labeled with Alexa Fluor-647-phalloidin (blue) and untethered actin filaments were labeled with rhodamine phalloidin (red). The movie capture began as soon as possible after initiation of the assay. As the assay proceeded, the individual myosin filaments coalesced into discrete stacks. The stacks were dynamic and could lose myosin filaments units as they moved along actin (*Video 7*). These phenomena suggests that the interaction forces holding filament units together are similar in magnitude to the force that a myosin filament exerts on actin, and that myosin filaments passing in close proximity are sufficient for stack generation.

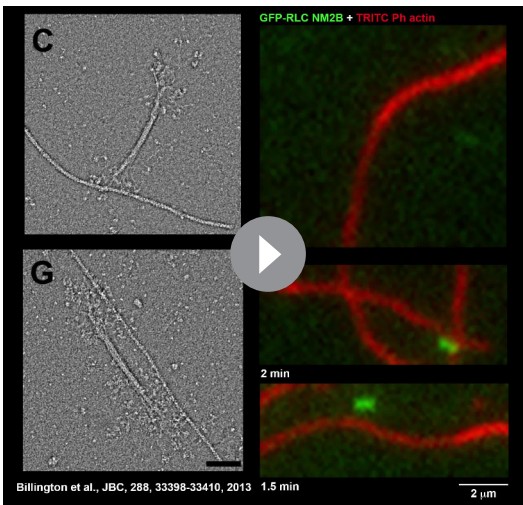

**Video 2.** Different modes of physical interactions between NM2-B filaments and actin during processive movements. Left panels are EM images of NM2-B filaments that interact with an actin filament using the myosin heads from one (upper panel) or both their ends (lower panel) (images taken from *Billington et al., 2013*). In the right panels, movies from single filament in vitro motility assays show several examples of GFP-RLC-NM2-B filaments (green) moving along actin filaments (red). It can be seen that NM2-B filaments were able to move with their long axis either perpendicular or parallel to the actin filaments according with the EM images. Moreover, upper and middle right panels show clearly GFP-RLC-NM2-B filaments that tumbled on actin filaments (arrows).
DOI: https://doi.org/10.7554/eLife.32871.009

## More than four NM2B motors are required for processive movement

The duty ratio of a single NM2-B motor (subfragment-one) is a function of the actin and ADP concentration but is likely to be at least 0.23 (*Wang et al., 2003*). The apparent duty ratio of the filament, $r_f$, is given by the following equation:

$$r_f = (1 - (1 - r)^n) \qquad (1)$$

where n = number of motor domains in the filament (*Nagy et al., 2013*). This equation assumes that each motor is capable of interacting with actin. In the presence of ATP, NM2-B molecules only interact with actin via one motor at a time (*Nagy et al., 2013*), so corrections for intrahead gating as shown by *Kovács et al. (2007)* do not have to be considered. The processive single molecule double-headed cargo motor, myosin 5a has a duty ratio of 0.9 calculated using the experimentally determined single-head duty ratio of 0.67 and *Equation (1)* (*De La Cruz et al., 1999*). Assuming that $r_f \geq 0.9$ is required to allow for processive movements, *Equation (1)* predicts that about nine NM2-B motors would need to be physically in a position to bind the actin filament to ensure that at least one of these motors is bound to actin at any given time. If no motors in a myosin filament are bound to actin, it would terminate a processive run and diffuse away from actin. Since there are, on average, 30 motors per half filament, it is not surprising that NM2-B filaments move processively as single units.

To test this hypothesis directly, we artificially reduced the number of motors in a filament by co-polymerization of full-length NM2-B with a nonmuscle myosin-2B tail fragment which was N-terminally truncated and lacked the motor and neck domains. Numerous studies have shown that tail fragments from various myosins II do not form discrete 300 nm bipolar assemblies, but instead polymerize into large irregular aggregates or paracrystalline arrays (*Cohen et al., 1970*; *Franke et al., 2005*). We replaced the motor and light chain binding domains at the N-terminus of the myosin NM2-B heavy chain with a HaloTag. This chimeric myosin tail fragment polymerized into bipolar filaments with roughly the same length and number of molecules as wild-type NM2-B filaments (*Figure 2A*, middle panel). The HaloTag moieties (MW = 33 kDa) can be seen as discrete globular domains projecting away from the filament backbone.

We co-polymerized full-length HaloTag-NM2-B-labeled with Alexa Fluor 488 (AF488) and HaloTag-NM2-B tail fragments labeled with tetramethylrhodamine (TMR). Changing the ratio of NM2-B molecules to tail fragments allowed us to titrate the number of motor domains present in these co-filaments (*Figure 2A*, right panel). We define $R_{NM2-B}$ to be the average fraction of NM2-B motor domains in a co-filament relative to 100% NM2-B filaments. Three different mixing ratios were used in our experiments, 1:1, 1:2 and 1:5 (NM2-B molecules:NM2-B tails) that correspond to $R_{NM2-B}$ values of 0.5, 0.33 and 0.17, respectively, if the two molecular species polymerized randomly into filaments. We used the co-filaments to estimate the minimum number of myosin motor domains that are required for the NM2-B filaments to be processive. For all three mixing ratios, co-filaments were observed moving processively along actin filaments. The run length and velocity of co-filaments at different mixing ratios were then determined (*Figure 2B and D*, *Videos 8–10*).

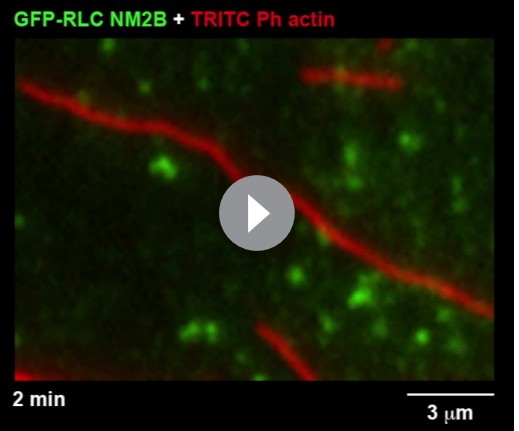

**Video 3.** A single myosin filament can interact simultaneously with two actin filaments. Left panel shows an EM image of an NM2 filament interacting with two actin filaments simultaneously (image taken from *Billington et al., 2013*). The movies in the right panels show examples of GFP-RLC-NM2-B filaments (green) that move along an actin filament (red) and stop at the intersection with another actin filament with opposite polarity (magenta arrows). In this situation, as in the EM image on the right, the NM2-B filaments interact with their bipolar ends with two different actin filaments in an isometric condition.
DOI: https://doi.org/10.7554/eLife.32871.010

**Video 4.** Actin filaments can be moved by myosin filaments bound to another actin filament. In this movie, an actin filament previously free in solution lands on a surface immobilized actin filament (blue arrow). The free actin filament is moved by an GFP-RLC-NM2-B filament that is bound to the immobilized actin filament. The NM2-B filament responsible for this movement is also translocating along the immobilized filament. Near the end of the movie, the free actin filament takes an abrupt turn as it is caught by a myosin filament that is bound to the surface close to the immobilized actin filament.
DOI: https://doi.org/10.7554/eLife.32871.011

The $R_{NM2-B}$ values indicated in the previous paragraph are theoretical values calculated assuming that the co-polymerization of the HaloTag-NM2-B tail fragments and the HaloTag-NM2-B full-length myosin molecules was unbiased. To define the actual values of $R_{NM2-B}$ for filaments that were moving along the actin filaments, we first determined the average fluorescence intensity of 100% AF488-HaloTag-NM2-B filaments moving on actin ($I_{2B}$). The average fluorescence intensity of co-filaments moving on actin was then measured in the same 488 nm excitation channel ($I_{2B,cof}$) (*Figure 2—figure supplement 1*). We calculated $R_{NM2-B}$ according to the following equation:

$$R_{NM2-B} = \frac{I_{2B,cof}}{I_{2B}} \tag{2}$$

We then calculated the average number of NM2-B motors domains per co-filament ($n$) according to *Equation 3*:

$$n = R_{NM2-B} \cdot n_c \tag{3}$$

where $n_c$ is the average number of motors present in filaments formed by NM2-B (=60) as estimated from EM studies (*Billington et al., 2013*). The values of $R_{NM2-B}$ and $n$ at each mixing ratio are reported in *Supplementary file 1* and *Figure 2—figure supplement 2A*. *Figure 2—figure supplement 2B* shows the relation between $n$ and mixing ratio. We found that the average percentage of full-length NM2-B in moving filaments was slightly higher than predicted for random association of the HaloTag-NM2-B tail fragment and the full-length myosin molecule. This was particularly evident for mixtures containing a large excess of tail fragment.

For each mixing ratio examined, the filament run length was fit to a single exponential decay (*Figure 2B*). The characteristic run length decreased as the number of motor domains decreased from 0.98 ± 0.03 µm, when there were on average 18 motors per half filament, to 0.45 ± 0.04 µm at an average of 9 motors per half filament (*Figure 2C*). The data in *Figure 2C* were fit to a linear equation. The intercept of this line with the $x$ axis is 4.1 ± 1.3 motors per half filament,

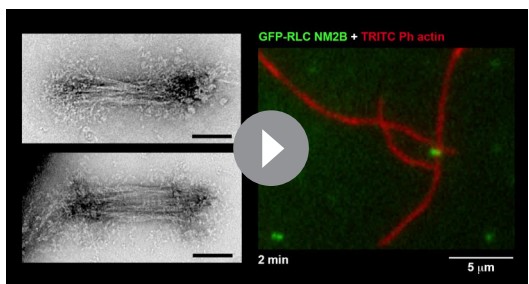

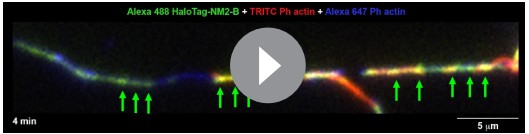

**Video 5.** Formation of myosin filament stacks. Left panel shows an EM images of GFP-RLC-NM2 stacks. These supramolecular structures are formed by lateral and serial interactions of single NM filaments. The movie in the right panel shows an object larger and brighter than a single filament, likely a stack of NM2-B filaments, that lands near an actin filament (blue arrow). A single NM2-B filament can be seen leaving the stack, while another joined it (magenta arrows).
DOI: https://doi.org/10.7554/eLife.32871.012

**Video 6.** Formation of myosin stacks. Actin filaments bound to the surface are labeled with Alexa-647-phalloidin (blue). Free actin filaments are label with Rhodamine phalloidin (Red). The myosin filaments are HaloTag-NM2-B labeled with AlexaFluor488 (green). At the start of the assay, numerous individual NM2-B filaments are bound to and moving along the fixed actin filament and free actin filaments can be seen to be moved by these same myosins. Whenever an individual myosin filament overtakes another the two filaments merge to form a stack until finally, only a small number of myosin filament stacks remain.
DOI: https://doi.org/10.7554/eLife.32871.013

suggesting that the minimum number of NM2-B motors that are required to maintain processive movement is at least 5. In contrast to the strong dependence of the run length on the number of NM2-B motors present in the co-filaments, the velocity of filament movement has little dependence on motor number (*Figure 2D,E*; and *Figure 2—figure supplement 2*).

## NM2-A filaments do not move processively under conditions where NM2-B filaments are processive

We next examined whether filaments formed of 100% NM2-A could move processively similar to our observations for NM2-B. Despite having similar numbers of motors/filament as NM2-B filaments, no processive movements of GFP-RLC-NM2-A filaments were observed under identical experimental conditions (*Video 11*). Strikingly, even binding events of the NM2-A filaments to the actin filaments were rarely observed. This was not due to lack of enzymatic or mechanical activity, since NM2-A monomers bound to a coverslip surface smoothly propelled actin filaments in the gliding assay (*Video 12*). We hypothesized that the absence of processive movement was not due to a lack of activity of the monomers, but rather an intrinsic difference in the kinetic properties of NM2-A versus NM2-B molecules. Note, that in contrast to a previous study (*Diensthuber et al., 2011*), we found no effect of phalloidin on the movement of actin filaments by NM2-A (*Supplementary file 3*).

## NM2-A filaments move processively in buffer approximating cellular viscosity

The environment that NM2 experiences in cells differs in several respects from our in vitro conditions explored thus far. The viscosity experienced by myosin filaments in cells is likely to be considerably higher than in the aqueous medium that was used for the in vitro motility studies (*Kalwarczyk et al., 2011*) and, in vivo, NM2 molecules can form heterotypic filaments that are composed of more than one myosin paralog (*Beach et al., 2014*; *Shutova et al., 2014*). Therefore, we first explored whether inclusion of 0.5% methylcellulose in the assay buffer, which gave a viscosity similar to that measured in the cytoplasm of cells for objects the size of NM2 filaments (see Materials and methods) (*Kalwarczyk et al., 2011*), would alter the motile properties of NM2-A and NM2-B filaments. Under these conditions, HaloTag-NM2-A filaments showed robust processive movement with an average velocity of $133 \pm 75$ nm $\cdot$ s$^{-1}$ (mean $\pm$ SD, n = 143) at 30°C (*Figure 3A–C* and *Video 13*). We were unable to measure an average run length since most of the NM2-A filaments moved to the end of the actin filaments or became stuck at actin-actin junctions (note vertical lines on kymographs in *Figure 3B*).

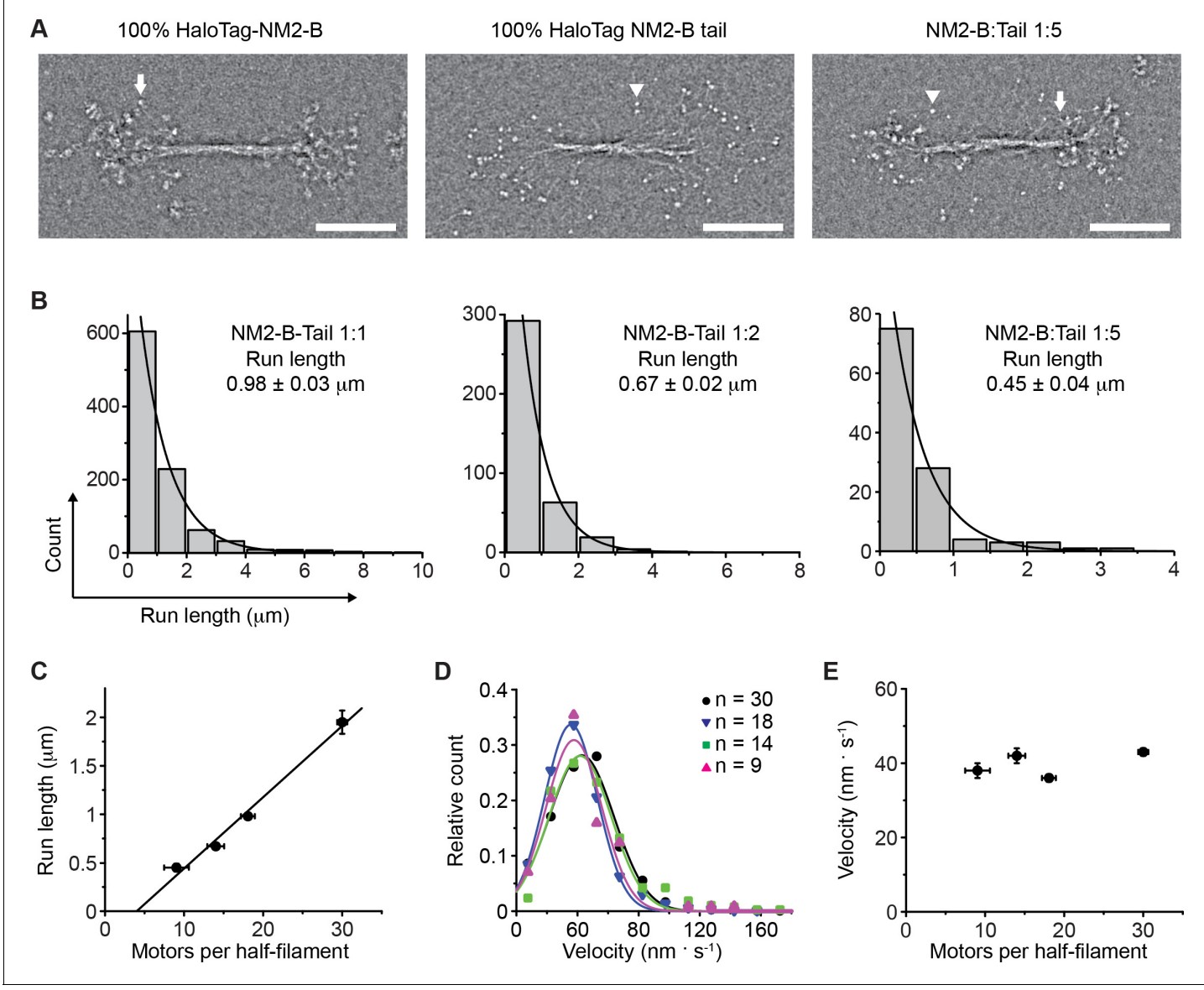

**Figure 2.** Determination of the number of NM2-B motor domains required for filament processivity. (**A**) Negative stain EM images of bipolar filaments formed by 100% HaloTag-NM2-B (left panel), 100% Halotag-NM2-B tail fragments (middle panel) and bipolar co-filament at 1:5 mixing ratio (NM2-B: Tail) (right panel). Arrows indicate a myosin motor domain. Arrowheads indicate a HaloTag moiety which can be seen as discrete globular domain smaller in size that the motor domain. (**B**) Frequency distribution histograms of NM2 co-filament run length. Black lines are the single exponential fit to the data. The mixing ratio and the characteristic run length obtained from the fit are indicated in each panel (n = 957, 378 and 113 for mixing ratios of 1:1, 1:2 and 1:5 NM2-B:Tail, respectively $R^2$ = 0.99 for each fitting). (**C**) Characteristic run length is plotted as a function of number of motors per half filament. The black line is the linear fit to the data ($R^2$ = 0.99). (**D**) Frequency distribution of velocity for all mixing ratios of co-filaments. The experimentally determined average number of motors per half filament are given for each mixing experiment in the inset. Lines are the Gaussian fits to the data. The velocity obtained from the fit is reported in *Figure 2—figure supplement 2*.. (**E**) The dependence of the characteristic velocity on the number of motors in a half filament. In all panels, errors represent S.E.M. and sample size is given above.

DOI: https://doi.org/10.7554/eLife.32871.015

The following source data and figure supplements are available for figure 2:

**Source data 1.** Data for run lengths of NM2-B and the hybrid filaments containing NM2-B and NM2-B tail fragment.
DOI: https://doi.org/10.7554/eLife.32871.019
**Figure supplement 1.** HaloTag-NM2-B and HaloTag-NM2-B myosin tail co-polymerize.
DOI: https://doi.org/10.7554/eLife.32871.016
**Figure supplement 1—source data 1.** Fluorescent intensity values of hybrid filaments of NM2-B and NM2-B trail fragment shown in Panel A.
DOI: https://doi.org/10.7554/eLife.32871.017
*Figure 2 continued on next page*

*Figure 2 continued*

**Figure supplement 2.** Rate of movement of hybrid filaments of full length NM2-B and NM2-B Tail.

DOI: https://doi.org/10.7554/eLife.32871.018

When 0.5% methylcellulose was used with HaloTag-NM2-B filaments, the processivity of the filaments greatly increased to the extent that virtually all NM2-B filaments moved the entire remaining length of the actin filament (*Video 14* and *Figure 3D,E*). Similar results were obtained with GFP-RLC-NM2-B where many of the myosin filaments remained bound to the ends of the actin filament (*Video 15*). This is particularly notable from the bright vertical lines in the kymograph indicative of multiple NM2-B filaments accumulating at the ends (*Figure 3E*). Similar behavior was described for smooth muscle myosin filaments in the presence of methylcellulose (*Haldeman et al., 2014*).

## Co-filaments of NM2-A and NM2-B move processively with intermediate motile properties determined by the proportion of each paralog

Given the strikingly different behaviors of NM2-A and NM2-B filaments described above, we next investigated how co-filaments containing both myosin paralogs moved in our assay. We co-polymerized tetramethylrhodamine (TMR)-HaloTag-NM2-A and AF488-HaloTag-NM2-B at three different ratios (2:1, 1:1 and 1:2; NM2-B:NM2-A) to form co-filaments. These mixing ratios should result in an average fraction of NM2-A molecules in the filaments ($F_{NM2-A}$) of 0.33, 0.5 and 0.67, respectively if the two paralogs co-polymerized randomly. Dual-wavelength analysis of NM2-A and NM2-B fluorescence intensities confirmed these filament compositions (*Figure 4—figure supplement 1* and *Supplementary file 2*) and demonstrated that the two paralogs did co-polymerize randomly. We studied the motility of the mixed isoform filaments using the single filament TIRF assay in the absence of methylcellulose (*Figure 4*, *Videos 16–18*). Since NM2-B is the slower, but the more processive of the two NM2 isoforms, the velocity and run length of 100% NM2-B filaments ($F_{NM2-A}$=0) previously measured represent the lower boundary for the velocity (*Figure 1C,D*) and the upper boundary for the run length (*Figure 4D*) of the mixed isoform filaments. The velocity for 100% HaloTag-NM2-A filaments ($F_{NM2-A}$=1) was measured in the presence of methylcellulose since processive movements for this myosin were otherwise not observed (*Figures 3C* and *4D*, red line and symbols and *Video 13*). Due to the lack of processivity of 100% NM2-A filaments in the absence of methylcelluose, run length was set to zero.

For all three mixing ratios, the filaments showed a robust processivity that allowed run length and velocity to be measured. The isoform composition of each moving filament was determined by measurement of the intensities of the two colors and varied only slightly from the ratios expected for random co-polymerization (*Figure 4—figure supplement 1*). We did not see evidence for the filaments changing their isoform composition during an actin-attached processive movement. Run length distributions were well described by a single exponential decay (*Figure 4A*) and decreased linearly with increasing $F_{NM2-A}$ from 1.88 ± 0.14 μm (mean ± S.E.M., n = 1121) at $F_{NM2-A}$ = 0.72, to 0.79 ± 0.01 μm (n = 1454) at $F_{NM2-A}$ = 0.24 (*Figure 4D*). The distributions of the velocity of the filaments at each mixing ratio were not well fit to a single Gaussian (*Figure 4—figure supplement 2*). This is particularly evident at the 1:2 mixing ratio. For this reason, we decided to calculate the arithmetic mean velocity of all the filaments at a given mixing ratio to represent *v*. The velocity of the mixed filaments is largely determined by the velocity of the slower moving NM2-B (*Figure 4C*). This was consistent with previous results from actin gliding in vitro motility assays, which show that slower myosins dominate the velocity when myosin 2 isoforms of different inherent velocity are randomly bound to the coverslip surface, which was modeled by assuming the two myosin isoforms affect the attachment lifetimes of each other (*Cuda et al., 1997*). Our results show that the mechanical properties of single heterotypic NM2 filaments can be varied depending on the ratio of their paralogs.

## Discussion

Our study has important implications for understanding how NM2 myosin filaments interact with actin filaments and function within the cell. Despite their almost identical structure, NM2-A and NM2-B filaments differ significantly in their motile and kinetic properties. NM2-B filaments have a

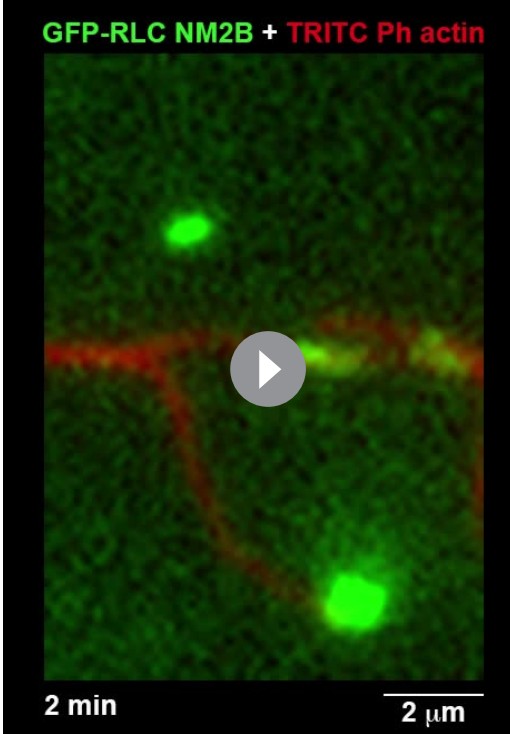

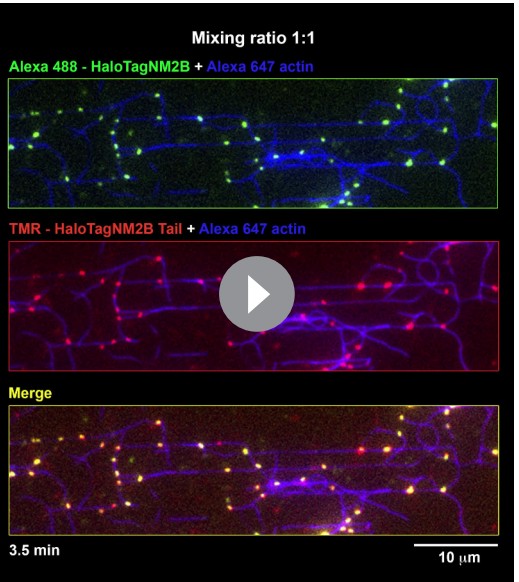

**Video 7.** Myosin filaments can dissociate from a myosin stack while moving. A bright stack of GFP-RLC-NM2B myosin filaments moving along an actin filament loses fluorescence intensity in quantal steps suggesting dissociation of one or more individual NM2-B units
DOI: https://doi.org/10.7554/eLife.32871.014

**Video 8.** Determination of the number of myosin motors required for processivity. The movies show co-filaments (yellow) of HaloTag-NB2B (green) and HaloTag-Tail (red) at a mixing ratio of 1:1 moving along surface immobilized Alexa-647-labeled actin filaments (blue). The NM2-B molecules and tail fragments were labeled with AlexaFluor488 and TMR, respectively. In these conditions, the average number of NM2-B motors per co-filament, $n$, was $36 \pm 2$.
DOI: https://doi.org/10.7554/eLife.32871.020

higher composite duty ratio and move slower than NM2-A filaments under the unloaded conditions of the single filament in vitro motility assay. The effect of force on the kinetics of NM2 paralogs has not been measured using optical trapping, but a previous transient kinetic study showed that the rate of dissociation of ADP from both acto-NM2-A and acto-NM2-B motor domains is affected by intramolecular strain that occurs when the two motors of a myosin molecule are bound to adjacent actin monomers in the presence of ADP (*Kovács et al., 2007*). However, this study revealed that NM2-B is significantly more strain dependent than NM2-A. Since the dissociation of ADP from acto-NM2 limits the rate of dissociation of NM2 from actin, this suggests that under strained conditions, NM2-B will have a significantly longer lifetime on actin filaments than NM2-A. This implies that NM2-B filaments might be better suited for maintaining tension on actin filaments in cells, whereas NM2-A would be more useful as a cargo motor for rapid but unsustained contractions or in the remodeling of the actin cytoskeleton, given its higher motility rate. In this regard, it is interesting that NM2-A is more prominent in the leading edge of cells where there are active actin dynamics, whereas NM2-B is typically more concentrated with stress fibers which are linked through adhesion complexes to generate forces on the substrate and where actin is highly dynamic (*Beach et al., 2014*). The longer attachment lifetimes and greater strain dependence of these attachments would allow NM2-B to maintain force more effectively and efficiently than NM2-A.

We also show in the present study that NM2-A and NM2-B molecules co-polymerize randomly in vitro consistent with reports that they readily form heterotypic filaments in cells (*Beach et al., 2014*; *Shutova et al., 2017*; *Shutova et al., 2014*). These co-filaments move processively over a range of NM2-A to NM2-B ratios. Our data show that co-filaments containing as few as six to eight molecules of NM2-B in the presence of a majority of NM2-A molecules continue to move processively, albeit

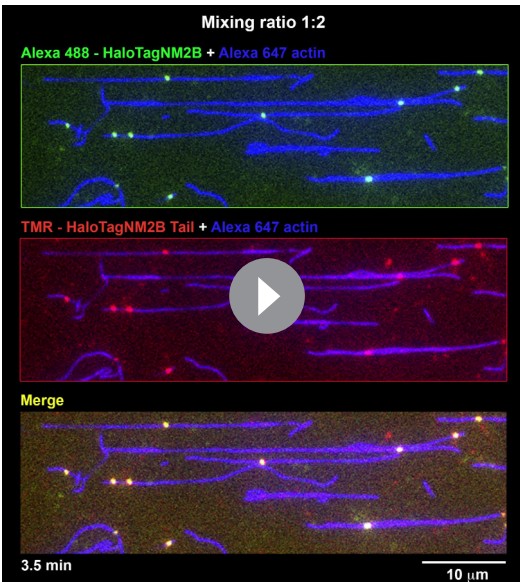

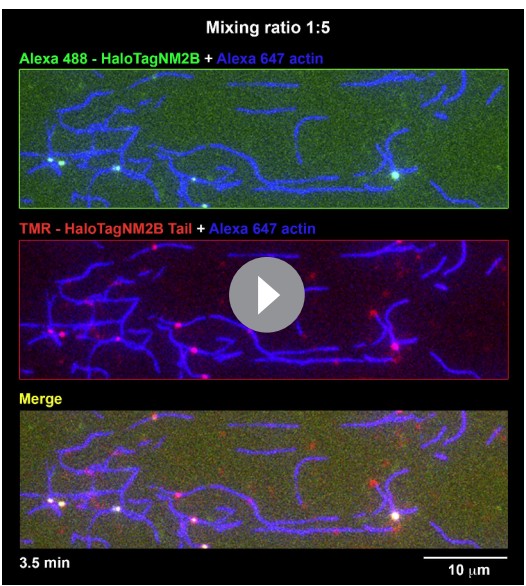

**Video 9.** Determination of the number of myosin motors required for processivity. The movies show co-filaments (yellow) of HaloTag-NB2B (green) and HaloTag-Tail (red) at a mixing ratio of 1:2 moving along surface immobilized Alexa-647-labeled actin filaments (blue). The NM2-B molecules and tail fragments were labeled with AlexaFluor488 and TMR, respectively. In these conditions, the average number of NM2-B motors per co-filament, *n* was 28 ± 2.
DOI: https://doi.org/10.7554/eLife.32871.021

**Video 10.** Determination of the number of myosin motors required for processivity. The movies show co-filaments (yellow) of HaloTag-NB2B (green) and HaloTag-Tail (red) at a mixing ratio of 1:5 moving along surface immobilized Alexa-647-labeled actin filaments (blue). The NM2-B molecules and tail fragments were labeled with AlexaFluor488 and TMR, respectively. In these conditions, the average number of NM2-B motors per co-filament, *n* was 18 ± 4.
DOI: https://doi.org/10.7554/eLife.32871.022

with shorter run lengths than filaments containing higher amounts of NM2-B. Interestingly, the velocity is strongly dictated by the slower NM2-B which is consistent with experiments using the sliding actin in vitro motility assay when fast and slow myosins are mixed on the surface (*Cuda et al., 1997*; *Harris et al., 1994*). These results imply that a cell can tune the mechanical output of a filament by varying its composite ratio of NM2 paralogs. Other sources of strain in a cell might affect the mechanical properties of NM2 paralogs. Strain could arise from two myosin filaments interacting with the same actin filament or from actin-binding proteins that might tether the filament to the cytoskeletal component or a membrane which might provide resistance to the NM2-directed motion.

By copolymerizing NM2-B with a tail fragment, we were able to determine that more than four motors need to be present at one end of a bipolar filament to ensure its processivity. This agrees remarkably well with calculations based on the duty ratio of the individual NM2-B motor domain (*Wang et al., 2003*). We also found that the rate of movement of the hybrid myosin filaments on actin were not significantly affected by the number of myosin motors available. These results suggest that the rate of movement of the myosin filaments are governed by the rate of detachment of myosin from actin. This is in contrast to a recent study with smooth muscle myosin filaments suggesting that the rate of filament movement is limited by the myosin attachment rate (*Brizendine et al., 2015*). There are several differences in the myosins and experimental setups that could have contributed to the different conclusions. First, smooth muscle myosin forms side-polar filaments in contrast to the bipolar filaments formed by NM2-B. Second, the duty ratio of smooth muscle myosin is low and it was necessary to include methylcellulose in the assay medium. In the presence of methycellulose, it might be possible for all the motors in a myosin filament to transiently be dissociated from actin without the myosin filament diffusing away before a new motor can bind and tether the myosin filament to actin. A second study from this same lab has recently been published containing a more elaborate model (*Brizendine et al., 2017*). This model allows that myosin filaments may move with

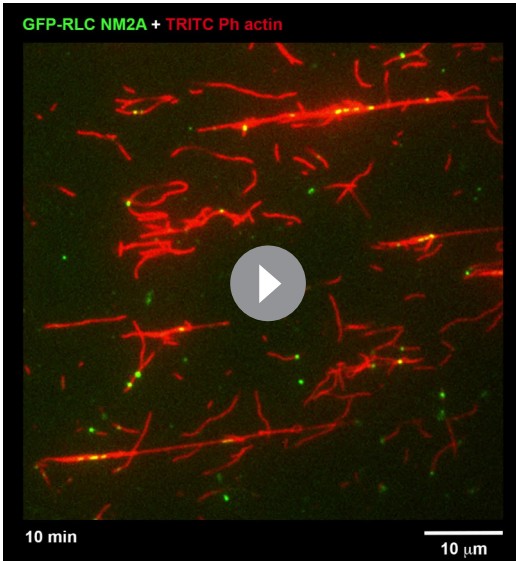

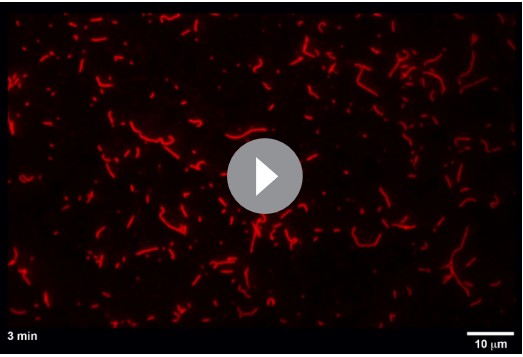

**Video 11.** NM2-A filaments do not move processively under conditions where NM2-B filaments do. This movie shows that under the same experimental conditions used to study the movement of NM2-B filaments, GFP-RLC-NM2-A filaments (green) were not able to move processively along actin (red) and even binding events were rarely observed. It must be noted that the experimental conditions and concentrations were the same as for NM2-B filaments experiments.
DOI: https://doi.org/10.7554/eLife.32871.023

**Video 12.** NM2-A monomers move actin filaments in the gliding actin in vitro motility assay. The movie shows actin filaments (red) moving as translocated by surface bound GFP-RLC-NM2-A monomers in an in vitro gliding assay. The actin filaments movement was smooth and continuous with only a small fraction of immobile filaments. This suggests that the absence of processive movement in the single filament TIRF experiments with GFP-RLC-NM2-A is not due to a lack of NM2-A activity.
DOI: https://doi.org/10.7554/eLife.32871.024

attachment or detachment limited kinetics depending on the myosin, myosin motor density and ionic conditions. The higher duty ratio of NM2-B would inherently bias the system toward a detachment limited mode. The observation that the rate of movement of the filaments did not vary with the number of contributing motors was consistent with a recent study which made synthetic myosin filaments mimics using DNA nanotubes where the spacing and density of myosin motor domains could be varied (*Hariadi et al., 2015*).

It is significant that tail fragments that were N-terminally fused with either the Halo-tag or a fluorescent fusion protein (data not shown) form bipolar filaments that, by negatively stained transmission electron microscopy, are similar to those formed by full-length molecules. In contrast, numerous studies have shown that myosin tail fragments that are not N-terminally capped by GFP or a HaloTag form much longer and thicker aggregates or paracrystals when polymerized either in vitro or when overexpressed in cells (*Cohen et al., 1970*; *Franke et al., 2005*). One potential use of capped tail fragments would be as a dominant-negative construct for myosin in cells where they would co-polymerize with full-length NM2 and thereby reduce the average number of myosin motors in a filament. However, the polymerization of these tail fragments lacking the light chain binding sites and therefore no RLC, would not be under the control of myosin light chain kinases and could result in inappropriate assembly and mislocalization of the myosin filaments in cells (*Beach et al., 2017*).

In view of the experimentally determined single molecule duty ratio of 0.05 for NM2-A, it is not surprising that these filaments failed to move processively in the absence of methycellulose. Calculations using *Equation (1)* show that more than 50 motors are required at one end of a NM2-A bipolar filament to ensure that at least one motor would be attached to an actin filament and ensure continuation of a processive run. There are only about 30 motors available at the end of a bipolar filament and not all of these are likely to be in a geometrically favorable position for actin filament interaction

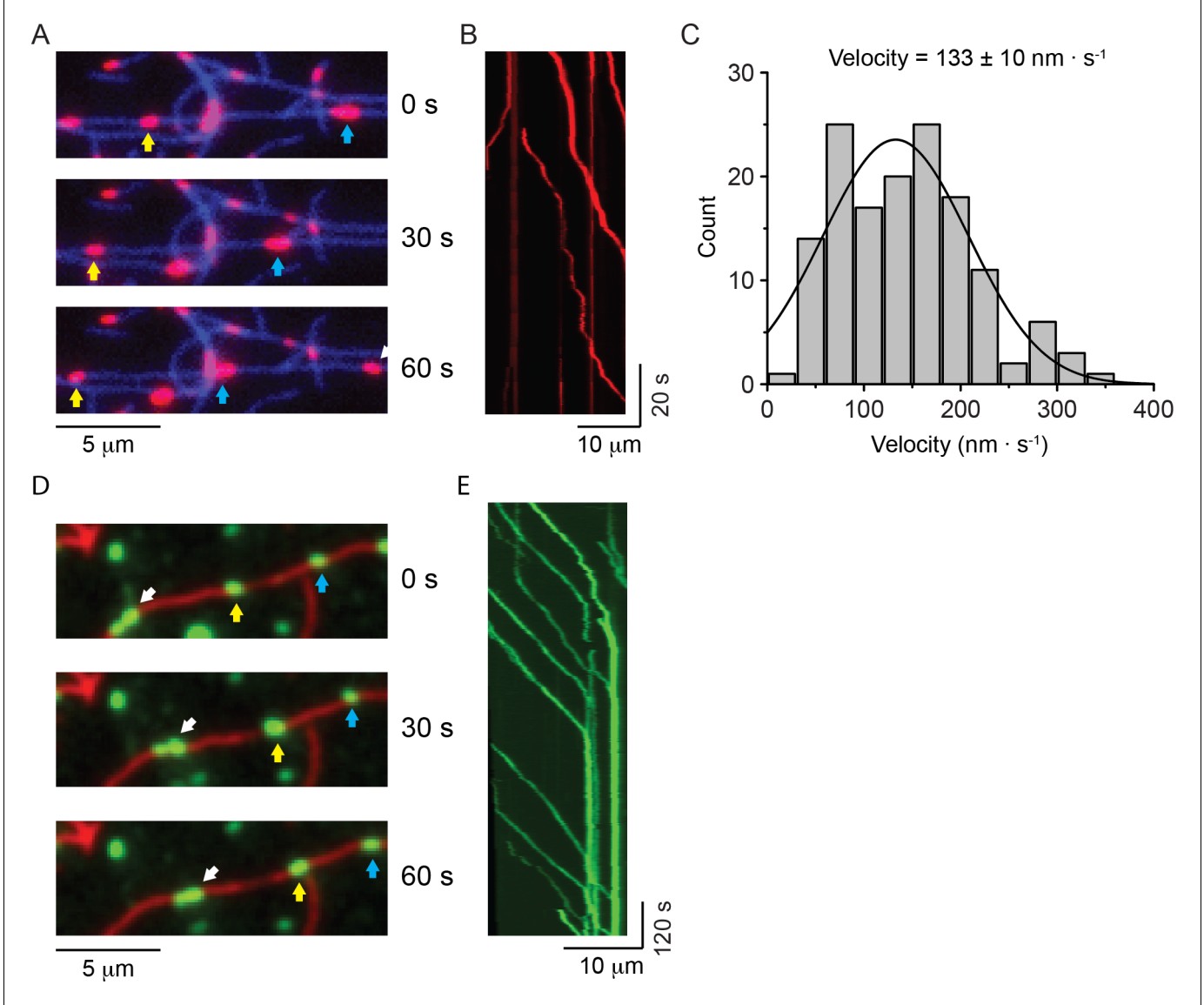

**Figure 3.** NM2-A filaments move processively in methylcellulose. (A) Movie frames showing HaloTag-NM2-A filaments (red, selected ones marked by blue and yellow arrows), moving along actin filaments (blue) in presence of 0.5% methylcellulose. Under these conditions, NM2-A filaments are able to move on actin for several microns without detaching. (B) Kymograph for NM2-A filaments in presence of 0.5% methylcellulose. (C) Frequency distribution histograms of NM2-A filament velocity (n = 143). The black line is the Gaussian fit to the data. The velocity is $133 \pm 10$ nm.s$^{-1}$ (mean ± SEM; SD = 75 nm.s$^{-1}$, $R^2$ = 0.81, ). (D) Movie frames showing HaloTag-NM2-B filaments (individual filaments marked by arrows), moving along actin filaments (green) in presence of 0.5% methylcellulose. In these conditions, the processivity of the filaments is increased dramatically relative to experiments in the absence of methylcellulose. (E) Kymograph for NM2-B filaments in presence of 0.5% methylcellulose. Most of the NM2-B filaments reached and accumulated at the end of the actin filaments as shown by the increasing in fluorescence intensity at the end of the actin filament and the vertical line at the end of the kymograph.

DOI: https://doi.org/10.7554/eLife.32871.025

The following source data is available for figure 3:

**Source data 1.** Velocities of NM2-A filament movement.

DOI: https://doi.org/10.7554/eLife.32871.026

at any given moment. Under these low viscosity conditions, the myosin filaments rapidly diffuse away from an actin filament when no motors were attached.

The viscosity experienced by objects the size of NM2 filaments in cells is expected to be between one and two orders of magnitude larger than that in a simple aqueous buffer. Therefore, we

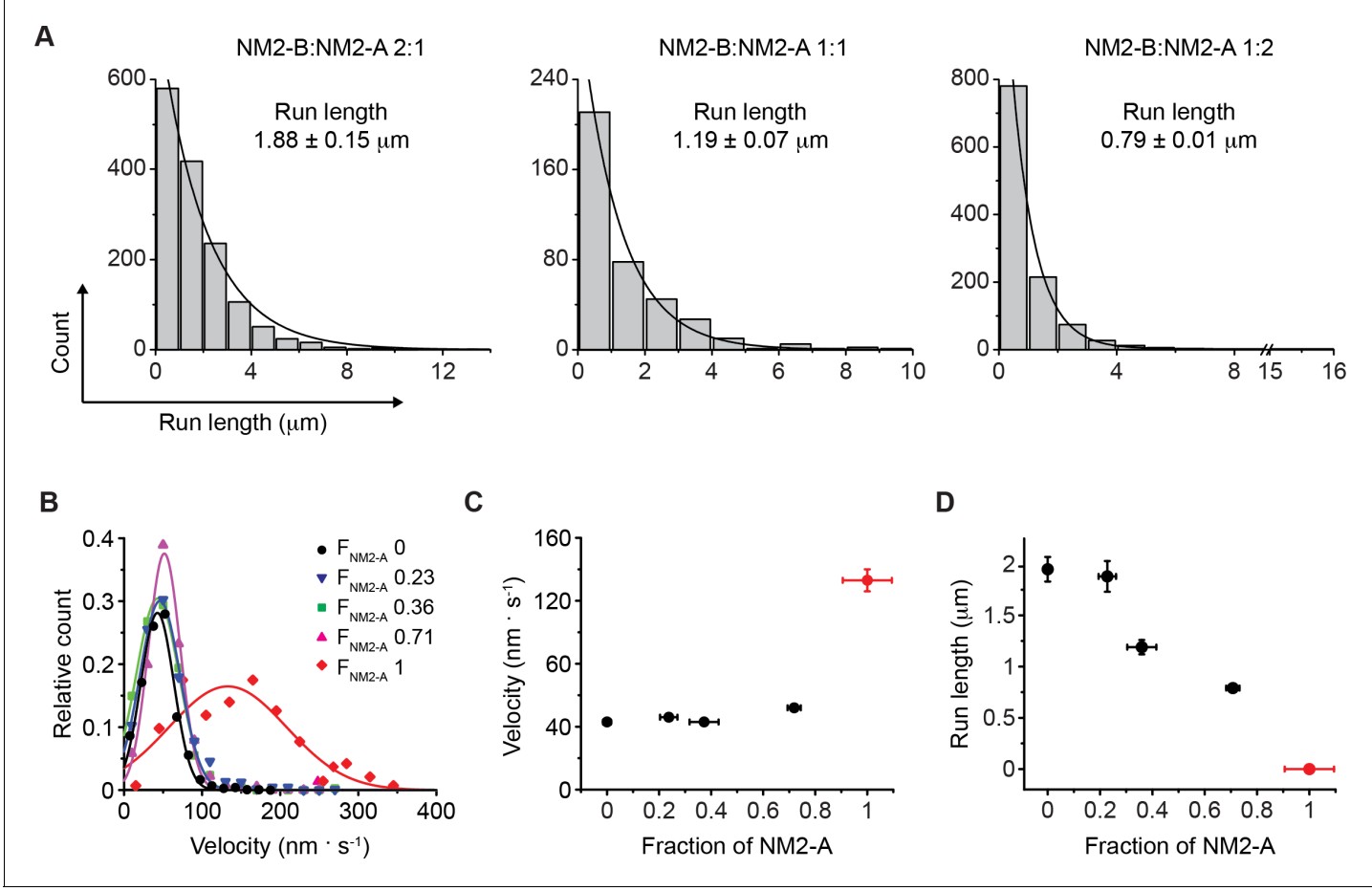

**Figure 4.** Motile properties of NM2-A and NM2-B co-filaments. (A) Frequency distribution histograms of the run length for NM2-A:NM2-B co-filaments. Black lines are the single exponential fit to the data. The mixing ratio and the characteristic run length obtained from the fit are indicated in each panel (n, $R^2$ = 1454, 0.98, 381, 0.99 and 1121, 0.99 for mixing ratios of 2:1, 1:1 and 1:2 NM2-B:NM2-A, respectively. The reported errors are the S.E.M. (B) Frequency distribution histograms of the velocity of mixed filaments for all mixing ratios. The $F_{NM2-A}$ (fraction of NM2-A in a filament) determined by quantification of the average NM2-A content per filament at each of the mixing ratios described in panel A is given in the insert The n values are 143, 1121, 381, 1454 and 1463 for $F_{NM2-A}$ of 1, 0.71, 0.36, 0.23, and 0, respectively. Lines are the Gaussian fit to the data. The distributions of these data are given in *Figure 4—figure supplement 2*. (C) The average velocity of co-filament movement as a function of $F_{NM2-A}$. The velocity for 100% NM2-A is depicted in red and comes from experiments conducted in the presence of methycellulose. The error bars are S.E.M. (D) Dependence of the characteristic run length on $F_{NM2-A}$. The run length decreases roughly linearly as fraction of NM2-A increases. The value for 100% NM2-A filaments, depicted in red, is set to 0 since NM2-A filaments do not move processively under these conditions. The error bars are S.E.M.
DOI: https://doi.org/10.7554/eLife.32871.027

The following source data and figure supplements are available for figure 4:

**Source data 1.** Run length data for NM2-A:NM2-B mixed filaments.
DOI: https://doi.org/10.7554/eLife.32871.031

**Figure supplement 1.** NM2-A and NM2-B co-polymerize into filaments.
DOI: https://doi.org/10.7554/eLife.32871.028

**Figure supplement 1—source data 1.** Fluorescent intensity values of mixed filaments of NM2-A and NM2-B.
DOI: https://doi.org/10.7554/eLife.32871.029

**Figure supplement 2.** Velocities of co-filaments of HaloTag-NM2-B myosin and HaloTag NM2-A moving on actin filaments.
DOI: https://doi.org/10.7554/eLife.32871.030

mimicked elements of the physiological viscosity by adding 0.5% methylcellulose to increase the viscosity by about 25-fold (*Kalwarczyk et al., 2011*). We show that under these higher viscosity conditions NM2-A filaments behave as processive units. A recent study of the motility of single filaments of smooth muscle myosin also found they did not move processively under low-viscosity buffer conditions, but that inclusion of methylcellulose into the assay chamber allowed for robust motility

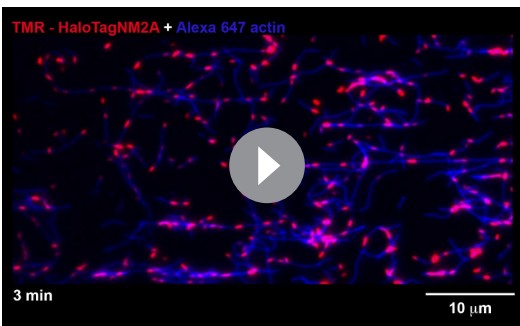

**Video 13.** NM2-A filaments are processive in the presence of methylcellulose. The movie shows TMR-labeled HaloTag-NM2-A filaments (green) moves processively along surface immobilized Alexa-647-labeled actin filaments (blue) in presence of 0.5% methylcellulose.
DOI: https://doi.org/10.7554/eLife.32871.032

(*Haldeman et al., 2014*). Methylcellulose was also used in in vitro motility experiments with skeletal muscle myosin-2 (*Murrell and Gardel, 2012*). Methylcellulose may act in two ways in this assay. By increasing solution viscosity, the diffusion rate of the NM2-A filaments is reduced sufficiently such that upon an instantaneous dissociation from actin, it would not diffuse away before another myosin motor could attach and restore processive movement. Methylcellulose is also a crowding agent which may influence the kinetics, and therefore the duty ratio of NM2-A (*Highsmith et al., 1996*).

Interestingly, inclusion of methylcellulose in assays of NM2-B filament produced a 'parking' behavior, where filaments were unable to detach upon reaching the end of the actin filament and where multiple myosin filaments accumulated at actin intersections. This behavior precluded a quantitative comparison of NM2-B filament run lengths under viscous and nonviscous conditions. The highly processive nature of NM2-B filaments is interesting to consider in that this myosin partitions to the rear of the cell and results in the production of an extended cell rear or tail, via the production of stable and long lived actomyosin structures (*Beach et al., 2014*; *Vicente-Manzanares et al., 2009*). The parking behavior of NM2-B filaments at high viscosity has also been observed for smooth muscle myosin filaments under similar assay conditions (*Haldeman et al., 2014*). The spatial self-sorting behavior of NM2-A and NM2-B filaments has recently been demonstrated to rely on a gradual enrichment of NM2-B in stress fibers during retrograde flow (*Beach et al., 2014*; *Shutova et al., 2017*). Our results offer an explanation for how NM2-B can become preferentially enriched towards the cell posterior in polarized cells, since NM2-B rich filaments have an enhanced activity to remain bound to actin and will thus tend to track rearwards with retrograde flow. Conversely, NM2-A rich filaments have a greater probability of detaching from actin and will thus undergo relatively high rates of turnover and recycling with respect to actin (*Shutova et al., 2017*).

Another significant difference in our in vitro assays and the cellular environment is that in our system each moving myosin filament typically has contact with only one actin filament. A single cluster of motor domains emanating from one end of a bipolar filament could potentially have interactions with about a 100 nm stretch along an actin filament (*Billington et al., 2013*). This length of actin encompasses around 40 actin monomers, many of which would be sterically unavailable to the myosin motor domains. Inside a cell, myosin filaments may interact with a bundle of actin filaments or with multiple individual actin filaments. Both of these cases would tend to increase the ability of myosin to processively interact with actin within its cellular environment. The ability of NM2 filaments to interact with actin in an 'end on' manner where the cluster of myosin motors on the opposite end of the bipolar filament project outward would also enhance the ability of NM2 filaments to cross-link and bundle actin filaments or to slide actin filaments relative to one another. The single filament in vitro motility assay will be very useful in dissecting this mode of actin-myosin interaction.

Recently, studies using super-resolution light microscopy found that individual myosin filaments align to form vertical stacks. The formation of these stacks appeared to require both actin and myosin dynamics (*Fenix et al., 2016*; *Hu et al., 2017*). We observed multi-filament structures that resemble these structures being formed when two NM2 filaments meet whilst moving on an actin filament. These structures are dynamic and can shed or add individual myosin filaments. Since dynamically treadmilling actin was not present in our in vitro assays, this suggests that merging and splitting of such structures is an intrinsic property of myosin filaments when they are brought into close proximity with each other. The physical basis for forming stacks is not known. The attraction is likely to be electrostatic since fewer filament stacks are observed by electron microscopy as the ionic strength is raised. To form the folded 'off' state, the two motor domains of a single myosin make an asymmetric interaction in which the loop 2 region of one motor interacts with the converter domain of another

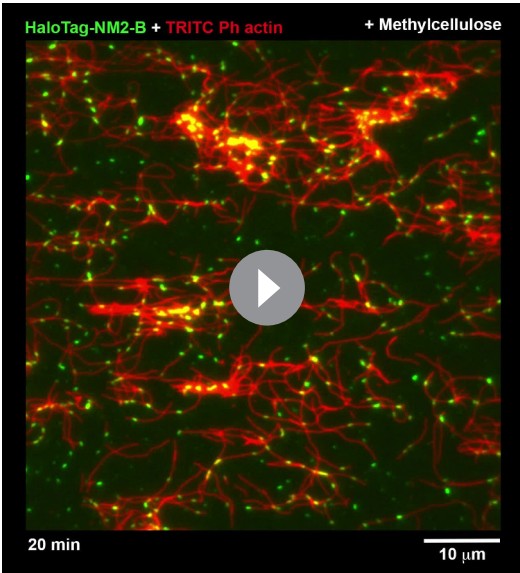

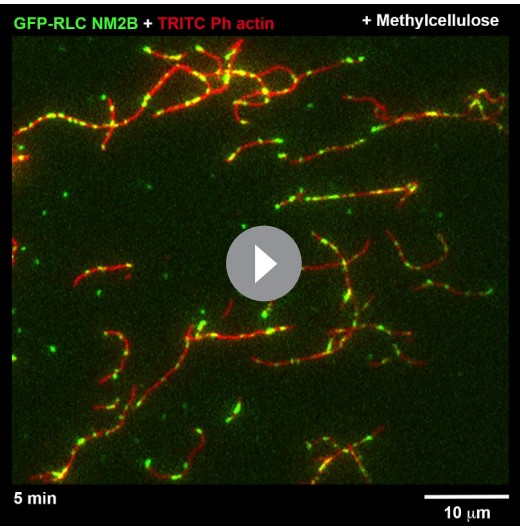

**Video 14.** NM2-B filament processivity is enhanced in the presence of methylcellulose. The movie shows Alexa-488-labeled HaloTag-NM2-B filaments (green) moving along surface immobilized actin filaments (red) in presence of 0.5% methylcellulose. It can be seen that the processivity of the NM2-B filaments was further increased over that seen in the absence of methylcellulose with many filaments moving greater than 10 μm. Many of the NM2-B filaments reached the end of the filament and did not detach.
DOI: https://doi.org/10.7554/eLife.32871.033

**Video 15.** NM2B filaments accumulate at the ends of actin filaments in the presence of methylcellulose. The movie shows GFP-RLC-NM2B (green) moving along surface immobilized actin filaments (red) in the presence of 0.5% methylcellulose. Note the strong accumulation of myosin at the ends of the actin filaments as the assay proceeds.
DOI: https://doi.org/10.7554/eLife.32871.034

(*Jung et al., 2008*; *Wendt et al., 2001*). This interaction appears to be broken upon phosphorylation of the regulatory light chains. It is possible that this weak interaction comes into significance when two filaments, each bearing 30 motors per filament end, come into close proximity. The fact that the stacks can disassemble suggests that the forces holding them together are not large.

The question remains as to why mammalian cells have three genes to encode nonmuscle myosins. Studies in mutant mouse models reveal that different nonmuscle myosin paralogs have different patterns of expression during development (for a review, see (*Ma and Adelstein, 2014*)). For example, in the early embryo, only NM2-A is expressed in the developing visceral endoderm (*Conti et al., 2004*). Ablation of NM2-A is lethal at embryonic day 6 since the cells cannot form appropriate adhesion complexes between cells. The DNA encoding the NM2-B heavy chain can be inserted into the NM2-A genetic loci resulting in expression of NM2-B during this early embryonic period. The NM2-B/NM2-A switch can rescue the early embryonic lethality, but the mouse experiences other defects later in development (*Wang et al., 2010*). Other studies reveal cases where paralog switching is not successful. NM2-A is specifically required for placental blood vessel formation and NM2-B is required for neuronal cell migration during brain development (*Ma et al., 2007*; *Wang et al., 2011*). In these cases, the opposite paralog cannot functionally substitute, suggesting that each of these myosins have features that are not redundant for some cellular functions.

Our results suggest that relating light microscopic observations of myosin filaments in cells to their mechanical output is not trivial. With the advent of super-resolution light microscopy, it is now possible to image single myosin filaments in cells, but most experimental methods used to date are not sufficient to truly understand the composition of the observed filaments (*Beach et al., 2017*; *Fenix et al., 2016*; *Hu et al., 2017*). If the investigator uses overexpression of a GFP-tagged RLC, then the heavy chain composition of the filament will be unknown. Similarly, visualization of myosin

filaments by overexpression of a GFP-tagged myosin heavy chain does not rule out the possibility that an observed filament could be a co-polymer with another NM2 paralog, or with myosin 18A (*MYO18*), which is an enzymatically inactive pseudo-myosin.

In summary, a cell has several potential mechanisms to mechanically fine tune or regulate the activity of a given myosin filament. These include forming heteromeric filaments composed of two or more NM2 paralogs (or myosin 18A) with differing mechanical properties and controlling the number of myosin molecules in a filament or the phosphorylation status of the regulatory light chain of the myosins in a filament (*Beach et al., 2017*; *Billington et al., 2015*). This means that for a given NM2 filament observed in a cell, there can be a wide range of mechanical properties which cannot be estimated by fluorescence intensity alone. It follows that that the behavior of individual NM2 filaments within a cell are highly unlikely to be uniform. Thus, increases in myosin filament density following a cellular perturbation or movement do not necessarily imply that the mechanical output will scale proportionally. These factors highlight the complexity of dissecting NM2 filament function in vivo. In this respect, the in vitro single filament motility assay is a sensitive way to provide information on the regulation of filament mechanical output and to inform future cell biological studies.

# Materials and methods

## Key resources table

| Reagent type (species) or resource | Designation | Source or reference | Indentifiers | Additional information |
|---|---|---|---|---|
| Cell line (Spodoptera fugiperda) | *Sf9* | Thermo Fisher Scientific | Thermo Fisher Scientific 11496015 | Maintained in Sf-900 III SFM |
| Recombinant DNA reagent | pFastBac1-NM2-A | PMID: 24072716 | N/A | |
| Recombinant DNA reagent | pFastBac1-NM2-B | PMID: 24072716 | N/A | |
| Recombinant DNA reagent | pFastBac1-NM2-B tail | This paper | NM2-B tail | Progenitor: pFastBac1-NM2-B |
| Software, algorithm | FAST | http://spudlab.stanford.edu/fast-for-automatic-motility-measurements | | |
| Software, algorithm | ImageJ | http://imageJ.nih.gov/ij | | |

## Generation of expression vectors

For the preparation of GFP-RLC-NM2 full-length molecules, the cDNA of full-length NM2-A and NM2-B MHCs were amplified and cloned into a modified pFastBac1 vector (FLAG-pFastBac1) encoding an N-terminus Flag-tag (DYKDDDK) for purification. An EGFP moiety was amplified and ligated to the N-terminus of the mouse regulatory light chain (EGFP-RLC) (*Kengyel et al., 2010*). The GFP-RLC cDNA was amplified and then cloned in a linearized pFastBac1 vector. GFP-RLC-NM2-A and NM2-B molecules were obtained by co-expressing FLAG-tagged MHCs with GFP-RLC and chicken essential light chain (ELC). For HaloTag-labeled NM2 molecules, the cDNA of a HaloTag moiety (IDT, Integrated DNA Technology) was ligated with a linearized FLAG-pFastBac1 plasmid (FLAG-HaloTag-pFastBac1) so that the plasmid encoded for an N-terminal FLAG-HaloTag moiety. The cDNA of NM2-A and NM2-B MHCs were amplified and then cloned in the FLAG-HaloTag-pFastBac1. HaloTag-NM2-A and -NM2-B MHCs were co-expressed with chicken RLC and ELC to obtain HaloTag NM2 full-length molecules. For the HaloTag-NM2-B tails, the cDNA encoding for residues 844–1976 of NM2-B MHC was PCR amplified and ligated with FLAG-HaloTag-pFastBac1 plasmid. In all cases, the HaloTag was fused to the N-terminus. Amplifications of cDNA and cloning of amplicons were performed using Primestar HR (Takara) and InFusion Technology (Clontech), respectively. The DNA sequence of cloned cDNA fragments were confirmed for all constructs by sequencing.

## Expression, purification and labeling of proteins

All constructs used in this paper were expressed using baculovirus/*Sf9* cells system (Invitrogen). Plasmid DNA was transformed into DH10-Bac *E. Coli* cells and recombinant bacmid isolated following manufacturer's protocols. First generation of baculovirus was generated by transfecting *Sf9* cells

with a mixture containing bacmid DNA and polyethylenimine (PEIMax, MW 40000; Polysciences) at a ratio of 1:9 in PBS buffer.

A baculovirus MOI of 3–5 were used to infect the *Sf9* cells for protein expression. Baculovirus infected *Sf9* cells were grown for 48–72 hr and harvested by sedimentation. Cell pellets were stored at −80℃. The proteins were purified according to previously published protocols (*Billington et al., 2013*; *Nagy et al., 2013*; *Wang et al., 2000*). Briefly, frozen pellets were thawed and homogenized using a ground glass homogenizer in buffer A (10 mM MOPS (pH 7.4), 5 mM MgCl$_2$, 0.1 mM EGTA) containing 0.5 M NaCl, 2 mM ATP, 0.1 mM phenylmethylsulfonyl fluoride and protease inhibitor cocktail (Roche). The proteins were purified by FLAG-affinity chromatography using M2 FLAG affinity gel (Sigma) and eluted in buffer A containing 0.5 M NaCl, and 0.5 mg/ml of FLAG peptide. The eluted proteins were dialyszed overnight in buffer A containing 25 mM NaCl and 1 mM DTT to induce the myosin polymerization into filaments. The protein pellet was then collected by centrifugation at 60,000 g for 30 min and dissolved in an appropriate amount of buffer A containing 0.5 M NaCl and 10 mM DTT. The purified proteins were analyzed using SDS-polyacrylamide gel (Invitrogen) followed by PageBlue staining (*Figure 1—figure supplement 1A*). Protein concentration was determined using a spectrophotometer and calculated according to the formula [myosin](mg/ml) = (A280 − A320)/εwhere ε, the extinction coefficient of the protein, was 0.52 mL·mg$^{-1}$·cm$^{-1}$ for myosins and 0.39 mL·mg$^{-1}$·cm$^{-1}$ for the NM2B tail (Online ExPASy ProtParam tool). The yield of the purification procedure was 1–2 mg of purified proteins. Myosin proteins were flash frozen in 20 µl aliquots and stored in liquid nitrogen until used. The GFP-RLC-labeled and HaloTag paralogs moved actin filaments in the sliding actin in vitro motility assay at the characteristic rate for the particular paralog (*Figure 1—figure supplement 1C*) and formed bipolar filaments that were of the same size and number of molecules as for wild-type myosins (*Figure 1—figure supplement 1B*).

Before the experiments, frozen NM2 solutions were thawed and phosphorylated using myosin light chain kinase (MLCK, 1–10 nM) in an overnight incubation on ice in buffer A containing 0.5 M NaCl, 0.2 mM CaCl$_2$, 0.1 µM calmodulin, 10 mM DTT and 1 mM ATP. HaloTag-NM2 molecules were fluorescently labeled together at the same time as the phosphorylation reaction by adding to the buffer a 10X excess of the chosen HaloTag-dye (Promega). To remove the excess dye, myosin was polymerized into filaments by dilution of the ionic strength to 25 mM with buffer A, centrifuged at 60,000 g for 45 min and then dissolved in buffer A containing 0.5 M KCl and 10 mM DTT. NM2 filaments were prepared by reducing the ionic strength of the solution to 150 mM with buffer A.

Skeletal muscle actin was purified from rabbit skeletal muscle (*Pardee and Spudich, 1982*). 10% biotinylated F-actin was prepared by polymerizing G-actin and biotinylated G-actin (Cytoskeleton) in KMEI buffer (50 mM KCl, 2 mM MgCl$_2$, 1 mM EGTA, 10 mM DTT and 10 mM MOPS (pH 7.4)). F-actin was labeled with fluorescent phalloidin (Thermo Fisher).

## Sliding actin in vitro motility assay

The sliding actin in vitro motility assay in which monomeric myosin is bound to the coverslip surface and the movement of fluorescently labeled actin filaments by this myosin is observed was conducted as described in *Sellers, 2006*). NM2-A bound to the coverslip surface was used in the experiments to determine whether phalloidin affects NM2-A motility. In one set of experiments, we separately added either Alexa-fluor-647-phalloidin-labeled actin filaments or filaments formed from actin monomers that were labeled with Atto-538 (Hypermol). In a second experiment preformed Alexa-Fluor-647-phalloidin-labeled actin filaments were mixed with preformed Atto-538-labeled actin filaments and then added to the in vitro motility flow chamber and the movement of both colored actin filaments were simultaneously measured.

## Single filament TIRF assay

The single filament TIRF assay was performed in flow cells made with a microscope slide (Corning Frosted Microscope Slides 75 × 25 mm; Thickness 0.9–1.1 mm), a coverslip (18 × 18 mm #1.5; Fisher Scientific) and double-side adhesive tape as previously described (*Kron and Spudich, 1986*). In the assay, biotinylated actin filaments were attached to the surface of the coverslip. In this study, two alternative protocols for the functionalization and passivation of the coverslips were used. In the first, 0.1% nitrocellulose was smeared and allowed to dry on the coverslip. The functionalized coverslip was then used to build the flow cell and a solution containing 5 mg·ml$^{-1}$ of biotinylated BSA was

applied to the flow cell. After 2 min, the flow cell was washed with motility buffer (20 mM MOPS (pH 7.4), 5 mM MgCl$_2$, 0.1 mM EGTA) containing 50 mM KCl and then a solution containing 1 mg·ml$^{-1}$ BSA was applied to the flow cell to reduce nonspecific interaction. After a second wash with 50 mM KCl motility buffer a solution containing 2 mg·ml$^{-1}$ NeutrAvidin was added to the flow cell. After 5 mins of incubation, the flow cell was washed again with 50 mM KCl motility buffer and it was ready for the binding of biotiny-lated actin filaments. Alternatively, the coverslip was treated with biotinylated Polyethylene glycol (Biotin-PEG) according to previously published protocols (*Breitsprecher et al., 2012*; *Haldeman et al., 2014*) with the following modi-fications. The coverslip were washed by sequen-tial sonication in 2% Hellmanex (Hellma GmbH and Co. KG), distillated water and 100% ethanol. In each step, the coverslips were sonicated for 10 min. After drying with N$_2$, the coverslip were plasma cleaned for 10 min using a Plasma system ZEPTO (Diener electronic, Germany). The con-centration of mPEG-silane (MW 2,000) and bio-tin-PEG-silane (MW 3,400) (both (Laysan Bio, Inc. Arab, AL) were 2 mg·ml$^{-1}$ and 10 μg·ml$^{-1}$, respectively. Prior to each experiment, a Biotin-PEG-treated coverslip was extensively rinsed with ddH$_2$O, dried with N$_2$ and used to build the flow cell. The flow cell was incubated with 10 mg·ml$^{-1}$ of BSA for 2 min. After washing

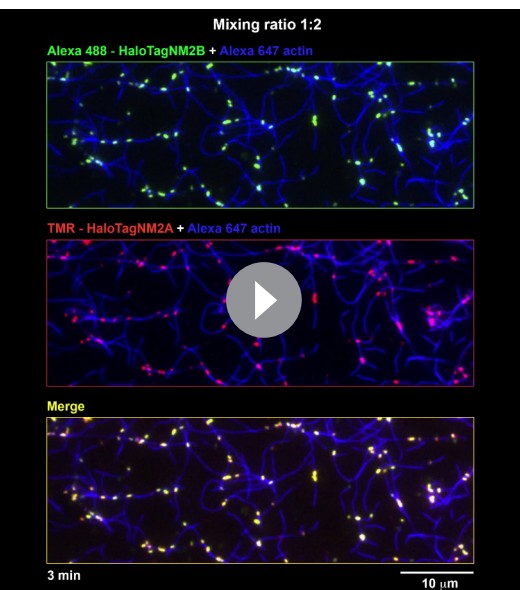

**Video 16.** NM2-A and NM2-B mixed filaments are processive. The movies show mixed paralog NM filaments moving along surface immobilized Alexa-647-labeled actin filaments (blue) at a mixing ratio of 2:1. HaloTag-NM2-B and HaloTag-NM-2A molecules were labeled with Alexa 488 (green) and TMR (red), respectively. In these conditions the actual average fraction of NM2-A molecules in the filaments, $F_{NM2-A}$, was $0.24 \pm 0.03$.

DOI: https://doi.org/10.7554/eLife.32871.035

with 50 mM KCl motility buffer a solution of 2 mg·ml$^{-1}$ NeutrAvidin was added and incubated for 5 min. The flow cell was then washed with 200 μl of 10 mg·ml$^{-1}$ BSA followed by 200 μl of 50 mM KCl buffer and it was ready for biotinylated actin binding. It is important to note that the Biotin-PEG treatment of the coverslips dramatically reduces the nonspecific interaction of NM2 filaments and the coverslip surface. Compare the number of myosin filaments stuck on the surface in *Videos 1* and *4*.

After functionalization and passivation of the coverslips phalloidin-labelled 10% biotinylated actin (200 nM) was added to the flow cell and incubated for 1 min. After washing with 200 μl of 150 mM KCl motility buffer the experiment was initiated by adding 30 μl of final buffer (150 mM KCl, 20 mM MOPS, 5 mM MgCl$_2$, 0.1 mM EGTA, 2 mM ATP, 10 mM DTT, 1–10 nM MLCK, 0.2 mM CaCl$_2$, 0.1 μM calmodulin, 25 mg·ml$^{-1}$glucose oxidase, 45 mg·ml$^{-1}$catalase, 2.5 mg·ml$^{-1}$ glucose) containing 30 nM NM2 in the form of myosin filaments. In the experiments where methylcellulose was used, after the binding of actin, a solution containing 30 nM of NM2 filaments in 150 mM motility buffer was added and incubated for 1 min. The NM2 filament motility was initiated by addition of 30 μl of final buffer with 0.5% methylcellulose into the flow cell. The change in order of addition of reagents compared to the experiments in the absence of methylcellulose was necessitated since free myosin filaments tended to form larger structures in the presence of methylcellulose. The viscosity of 0.5% methylcelluose (~25 mPa·s according to manufacturer's technical sheet (http://www.sigmaaldrich.com/content/dam/sigma-aldrich/docs/Sigma/Product_Information_Sheet/2/m0512pis.pdf)) was cho-sen to match the viscosity within cells for large objects such as NM2 filaments (24–44 mPa·s) (*Kalwarczyk et al., 2011*).

Movies of the NM2 filaments moving on actin were collected on an inverted Nikon Eclipse Ti-E microscope with an H-TIRF module attachment, a CFI60 Apochromat TIRF 100X Oil Immersion Objective Lens (N.A. 1.49, W.D. 0.12 mm, F.O.V 22 mm) and an EMMCD camera (Andor iXon Ultra 888 EMCCD, 1024 × 1024 array, 13 um pixel). The excitation light source was a Nikon LU-N4 Laser

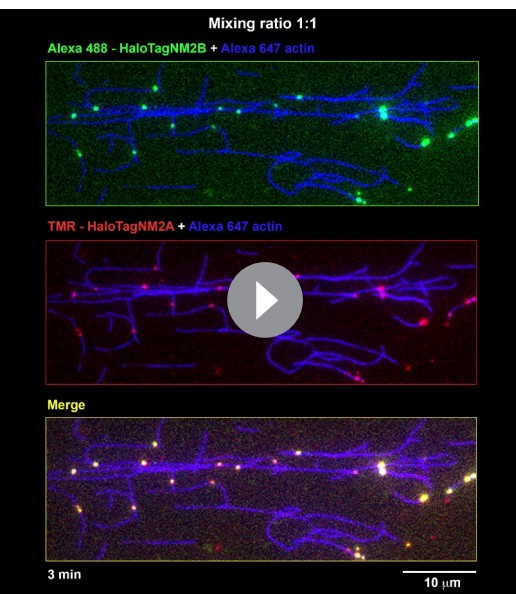
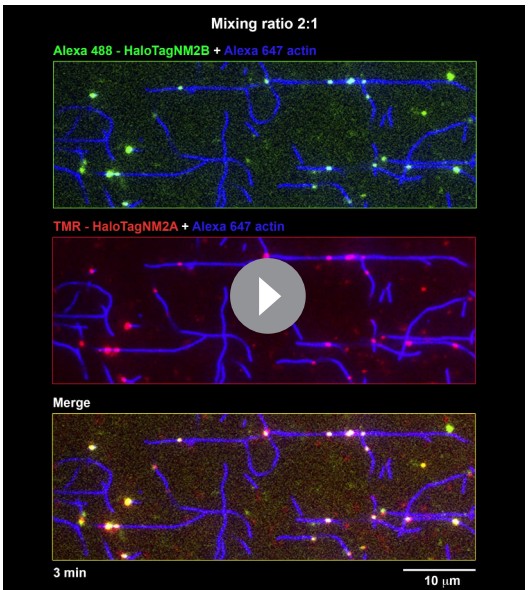

**Video 17.** NM2-A and NM2-B mixed filaments are processive. The movies show mixed paralog NM filaments moving along surface immobilized Alexa 647 labelled actin filaments (blue) at a mixing ratio of 1:1. In these conditions $F_{NM2-A}$ was $0.37 \pm 0.06$.
DOI: https://doi.org/10.7554/eLife.32871.036

**Video 18.** NM2-A and NM2-B mixed filaments are processive. The movies show mixed paralog NM filaments moving along surface immobilized Alexa-647-labeled actin filaments (blue) at a mixing ratio of 1:2. In these conditions $F_{NM2-A}$ was $0.72 \pm 0.03$.
DOI: https://doi.org/10.7554/eLife.32871.037

Unit equipped with four lasers (405 nm, 488 nm, 561 nm, 640 nm). The microscope was also equipped with an externally triggered fast wheel for the emission filters (HS-625 Filter Wheel, Finger Lakes Instrumentation (FLI), LIMA, NY) to increase the acquisition rate and at the same time reduce the crosstalk between channels. A stage top incubator (TOKAI HIT, Japan) was used to control the temperature of the flow cell.

Movies of NM2 filament movement lasting 5–10 min were recorded and, depending on the experiments, two (488 nm and 561 nm) or three (488 nm, 561 nm and 640 nm) channels were simultaneously acquired. For each movie shown in this paper the acquired channels, time and frame of acquisition, exposure time and temperature are indicated in the captions.

## Data analysis

Before the movies were analyzed, background subtraction was performed as follows. For each fluorescence channel, 20 images of the field of view were captured using the same laser power and exposure time used for the experiments. The background images from each channel were then averaged and the resulting image was subtracted from the respective channel of the movies. Each movie was analyzed individually for tracking of fluorescently labelled NM2 filaments using the TrackMate plug-in for ImageJ (National Institutes of Health). The TrackMate plug-in settings used for the analysis of the movies were the following: detector: LoG detector (estimated blob diameter: 1 µm, Threshold: 50–200), initial threshold: not set, view: HyperStack Displayer, tracker: Simple LAP tracker (Linking max distance: 1 µm, gap-closing max distance: 0.5 µm, gap-closing max frame gap: 2), filters on the tracks: Track displacement (above 150 nm). Each track identified by the software using these settings was manually checked using the Trackscheme displayer tool. Only myosin filaments that were observed to bind to actin, move and dissociate before reaching the end of the actin filament were included in the analysis. NM2 filaments that reached the end of the actin and dissociated were excluded from the analysis. Thus, we are measuring a minimum estimate of the run length. For each manually selected filament, the total fluorescence intensity per frame of each filament was calculated from the average intensity per pixel and spot size provided by the TrackMate analysis. The average total fluorescence intensity of each filament, *I*, was then calculated averaging the total fluorescence intensity for the entire track.

The fluorescence channels used for filament tracking varied according to the fluorophore used to label the NM2 filaments. For the experiments in which two fluorophores were used to label the molecules that form the NM2 filaments, filament tracking with TrackMate plug-in was performed using the 488 nm channel to measure the average velocity and run length and to calculate the intensity (I) of the Alexa Fluor 488 (AF488)-labeled molecules that form each filament. For the experiments with mixed isoform filaments, *I* of tetramethylrhodamine (TMR)-labeled molecules was instead calculated using a custom-written macro in ImageJ where the XY positions of the AF488 spots detected by TrackMate are used to draw a 1 μm squared ROI in the 561 nm channel around each filament. Then, for each ROI the macro calculated the total fluorescence intensity in each frame. *I* of TMR-labeled molecules that form each filament was then calculated averaging the total intensity of the respective ROI for the entire track.

To determine the actual fraction of NM2-A ($F_{NM2-A}$) in the co-filaments which were moving along actin in *Figure 4* and *Videos 16–18* we calculated at each mixing ratio the average total fluorescence intensity of TMR-labeled NM2-A molecules ($I_{2A,mix}$) and AF488-labeled NM2-B molecules ($I_{2B,mix}$) that form the filaments moving on actin. $I_{2A,mix}$ and $I_{2B,mix}$ are proportional to the number of NM2-A and NM2-B molecules, respectively, and $F_{NM2-A}$ was calculated according to the following equation:

$$F_{NM2-A} = \frac{\frac{I_{2A,mix}}{I_{2A,c}}}{\frac{I_{2A,mix}}{I_{2A,c}} + \frac{I_{2B,mix}}{I_{2B,c}}} \qquad (4)$$

where $I_{2A,c}$ and $I_{2B,c}$ are the average total fluorescence intensities of the 100% TMR-NM2-A filaments and 100% Alexa488-NM2-B filaments, respectively. The values of $F_{NM2-A}$ at each mixing ratio

## Electron microscopy

Full-length myosin and myosin tail fragments were mixed in buffer A containing 0.5 M NaCl to give the final molar ratios indicated in the text. The ionic strength was lowered by dilution into 10 mM MOPS (pH 7.0), 0.1 mM EGTA, 2 mM $MgCl_2$ and the required concentration of NaCl such that the final NaCl concentration was 150 mM and the final myosin concentration (full-length plus tail fragment) was 100 nM. Samples were incubated for 30 min on ice prior to making EM grids. A 3 μl drop of sample was applied to UV-treated carbon-coated copper grids and stained with 1% Uranyl Acetate (45 min UV treatment using a type R51 UV lamp with 5 cm between the bulb and grid surface (UV Products, Pasadena, CA)). Micrographs were recorded on a JEOL 1200EX II microscope operating at room temperature. Data were recorded on an ATM XR-60 CCD camera.

## Acknowledgements

We thank the NHLBI Light Microscopy Core Facility and the NHLBI Electron Microscopy Core Facility for the use of their microscopes, Fang Zhang for technical assistance with actin purification and Earl Homsher for reading the manuscript and providing comments. We thank Keir Neuman for discussion on the statistical treatment of the data. Funds from the NHLBI/NIH Intramural Research Program HL1001786 to JRS and from the NIDCD/NIH Intramural Research Program DC000039 to TBF supported this project.

## Additional information

### Funding

| Funder | Grant reference number | Author |
|---|---|---|
| National Institute on Deafness and Other Communication Disorders | DC000039 | Thomas B Friedman |
| National Heart, Lung, and Blood Institute | HL001786 | James R Sellers |

The funders had no role in study design, data collection and interpretation, or the decision to submit the work for publication.

## Author contributions

Luca Melli, Conceptualization, Resources, Software, Formal analysis, Investigation, Visualization, Methodology, Writing—original draft, Writing—review and editing; Neil Billington, Conceptualization, Software, Formal analysis, Investigation, Visualization, Methodology, Writing—review and editing; Sara A Sun, Methodology; Jonathan E Bird, Resources, Writing—review and editing; Attila Nagy, Resources, Methodology, Writing—review and editing; Thomas B Friedman, Resources, Supervision, Writing—review and editing; Yasuharu Takagi, Resources, Formal analysis, Validation, Investigation, Writing—review and editing; James R Sellers, Conceptualization, Data curation, Supervision, Funding acquisition, Investigation, Writing—original draft, Project administration, Writing—review and editing

## Author ORCIDs

Luca Melli (iD) http://orcid.org/0000-0002-8733-047X
Neil Billington (iD) http://orcid.org/0000-0003-2306-0228
Jonathan E Bird (iD) http://orcid.org/0000-0001-5531-8794
Thomas B Friedman (iD) http://orcid.org/0000-0003-4614-6630
James R Sellers (iD) http://orcid.org/0000-0001-6296-564X

## Decision letter and Author response

Decision letter https://doi.org/10.7554/eLife.32871.045
Author response https://doi.org/10.7554/eLife.32871.046

## Additional files

### Supplementary files

• Supplementary file 1. Results of mixing full-length NM2-B molecules with NM2-B tail fragments.
DOI: https://doi.org/10.7554/eLife.32871.038

• Supplementary file 2. Results of mixing NM2-A with NM2-B.
DOI: https://doi.org/10.7554/eLife.32871.039

• Supplementary file 3. Effect of phalloidin on the rate of gliding of actin by NM2-A.
DOI: https://doi.org/10.7554/eLife.32871.040

• Supplementary file 4. Data for the rate of actin filament gliding for actin labeled at Cys 374 with Atto 538 and with Alexa 647 phalloidin.
DOI: https://doi.org/10.7554/eLife.32871.041

• Transparent reporting form
DOI: https://doi.org/10.7554/eLife.32871.042

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
