## [Decision Letter]

Thank you for submitting your article "Bipolar filaments of human nonmuscle myosin 2-A and 2-B have distinct motile and mechanical properties" for consideration by *eLife*. Your article has been reviewed by three peer reviewers, and the evaluation has been overseen by Anna Akhmanova as the Senior Editor and Mohan Balasubramanian as the Reviewing Editor. The following individual involved in review of your submission has agreed to reveal their identity: Robert Anthony Cross (Reviewer #1).

The reviewers have discussed the reviews with one another and the Reviewing Editor has drafted this decision to help you prepare a revised submission.

The referees and the editors appreciated your careful study looking at copolymers containing myoIIA and myoIIB and deciphering their biophysical parameters. The referees believe that, with some revision, this paper will make a fine contribution to *eLife*, which will be of value to the molecular motor and the cell biology community.

Essential revisions:

A large number of minor points have been raised and I would like you to consider and address all of them, since nearly all of these are fixed by rewriting or by one experiment (the phalloidin bound actin inhibiting motile properties of myosin IIA).

1) The paper needs to clearly articulate how this work might influence thinking in the cell biology community and how it may inform various cell behaviours.

2) The general principle that variations in myosin isoform composition produce different properties is well-established in muscle myosin thick filaments. This work establishes that the blend of isoforms also determines the velocity and processivity of the much smaller myosin filaments that assemble in nonmuscle cells. The authors clearly feel that the nonmuscle myosin field has neglected the importance of isotype-mixing in determining mechanical output, but their data do not actually demonstrate that filaments in vivo vary in their mechanical output – first because they are not working in vivo, and second because there are no force measurements in this paper, just measurements of zero-load processivity. I recommend toning down the language about mechanical output and concentrating on velocity and processivity, whilst discussing the potential for force-generating and force-holding ability of NM2 filaments in vivo to vary.

3) Please clarify in the text whether the duty ratios given assume that the two heads of one molecule interact independently with actin. “[…] the duty ratio of an NM2-B monomer" – does this mean one head?

4) The Discussion considers the potential for filaments with different isotype mixes to deal differently with strain, but again, since there are no measurements of strain here, I think the authors should restrict themselves to pointing out the possibilities, rather than implying that the current results speak directly to the role of strain versus isotype-mix. The Kovacs (2007) study was about internal strain (strain between heads of one molecule). The internal stress and strain on myosins in filaments will be different in a nontrivial way (Hariadi, Nat Nano, 2015). External strain (for example, that coming from another myosin filament pulling on the same actin, could also be relevant in vivo.

5) Please comment, if possible, on whether exchange of fluorescent molecules between filaments occurs on a relevant timescale.

6) A paper by Diensthuber et al., (2011) suggests that phalloidin stabilization of actin inhibits the motile properties of NM2-A, but not NM2-B. Phalloidin is used in these experiments, so the authors should address this point about possible inhibition of the motor.

7) The Materials and methods do not reveal how the authors deal with limitation of actin filament length in calculating the run length.

8) The meaning of uncertainties are not always clear in the figures. For example, Figure 1 shows a distribution with a mean +/- 1 nm. I assume this is S.E.M. I would actually prefer to see the STD of the distributions throughout the manuscript, since that is the more useful parameter when comparing the different distributions. For example, the difference in the widths of the distributions between Figure 1 and Figure 3 is interesting.

9) Figure 2, third panel: It is difficult to believe that there can be confidence in the exponential fit of these binned data. Is this binning actually what was used to obtain the fit? Fitting to smaller bins, cumulative distributions, or via MLE would be more appropriate.

10) The rates of filament motility appear to be limited by the rate at which myosins detach from actin (Figure 2), which is at odds with the results of Brizendine et al., (2015), which suggest an attachment limited model. It would be helpful to the myosins specialists if this point was discussed.

11) The finding that myosin filaments can form stacks in vitro is interesting. Can the authors speculate on the mechanism by which the stacks form?

12) Subsection “More than five NM2B motors are required for processive movement”: The authors' discussion of the duty ratio does not consider head-head gating. Although this is mentioned later in the manuscript, it is appropriate to mention that this equation does not take this process into account.

13) Discussion, last two paragraphs: The wording in the paragraphs is awkward. The topic sentence of the first paragraph is: "Our results raise experimental concerns in elucidating the role of NM2s in cell biological experiments." However, it is not clear from the paragraph which concerns were raised by the results. The information in the second paragraph actually seems to be more related to this sentence.

---

## [Author Response]

Essential revisions:A large number of minor points have been raised and I would like you to consider and address all of them, since nearly all of these are fixed by rewriting or by one experiment (the phalloidin bound actin inhibiting motile properties of myosin IIA).1) The paper needs to clearly articulate how this work might influence thinking in the cell biology community and how it may inform various cell behaviours.

We attempted to do this in the original manuscript. Examples of where we had addressed this in the original manuscript include:

Introduction: “These data suggest that NM2-B is more adapted to a role of maintaining a high static tension level in cells, whereas NM2-A is geared to more rapid motile activity. […] Not all NM2 filaments have identical mechanical properties.”

Discussion: “This implies that NM2-B filaments might be better suited for maintaining tension on actin filaments in cells whereas NM2-A would be more useful as a cargo motor, for rapid but unsustained contractions or in the remodeling of the actin cytoskeleton, given its higher motility rate. […] The longer attachment lifetimes and greater strain dependence of these attachments would allow NM2-B to maintain force more effectively and efficiently than NM2-A.”

Discussion: “These results imply that a cell can tune the mechanical output of a filament by varying its composite ratio of NM2 paralogs.”

Discussion: “The highly processive nature of NM2-B filaments is interesting to consider in that this myosin partitions to the rear of the cell and results in the production of an extended cell rear or tail, via the production of stable and long lived actomyosin structures (Beach et al., 2014; Vicente-Manzanares et al., 2009).”

Discussion: “Our results offer an explanation for how NM2-B can become preferentially enriched towards the cell posterior in polarized cells, since NM2-B rich filaments have an enhanced activity to remain bound to actin and will thus tend to track rearwards with retrograde flow. Conversely, NM2-A rich filaments have a greater probability of detaching from actin and will thus undergo relatively high rates of turnover and recycling with respect to actin (Shutova et al., 2017).”

Discussion: “The ability of NM2 filaments to interact with actin in an “end on” manner where the cluster of myosin motors on the opposite end of the bipolar filament project outward would also enhance the ability of NM2 filaments to cross-link and bundle actin filaments or to slide actin filaments relative to one another.”

Discussion: “Since dynamically treadmilling actin was not present in our in vitro assays this suggests that merging and splitting of such structures is an intrinsic property of myosin filaments when they are brought into close proximity with each other.”

Discussion: “Thus, a cell has several potential mechanisms to mechanically fine tune or regulate the activity of a given myosin filament. […] In this respect, the in vitro single filament motility assay is a sensitive way to provide information on the regulation of filament mechanical output and to inform future cell biological studies.”

2) The general principle that variations in myosin isoform composition produce different properties is well-established in muscle myosin thick filaments. This work establishes that the blend of isoforms also determines the velocity and processivity of the much smaller myosin filaments that assemble in nonmuscle cells.

We now acknowledge in the revised manuscript that the different properties of skeletal muscle myosin isoforms have been known for some time (see review by Bottinelli and Reggiani (Bottinelli and Reggiani, 2000) from experiments studying unloaded shortening of single muscle fibers and we have now included a reference to an excellent review of this:

“It has long been known that the different skeletal muscle myosins-2 have distinct enzymatic and mechanical properties from studies in muscle fibers and of isolated proteins (Bottinelli and Reggiani, 2000).”

The authors clearly feel that the nonmuscle myosin field has neglected the importance of isotype-mixing in determining mechanical output, but their data do not actually demonstrate that filaments in vivo vary in their mechanical output – first because they are not working in vivo, and second because there are no force measurements in this paper, just measurements of zero-load processivity. I recommend toning down the language about mechanical output and concentrating on velocity and processivity, whilst discussing the potential for force-generating and force-holding ability of NM2 filaments in vivo to vary.

We do agree that we feel the difference in the properties of the nonmuscle myosin isoforms have been largely neglected in the motility/cytoskeletal field. The best example of this is that many cell biologists routinely use expression of a GFP-labelled RLC to follow “nonmuscle myosins”. While no one has really studied it quantitatively with pull downs, etc., I think we would all be very surprised if this light chain didn’t bind to each of the three NM2 isoforms in cells. Therefore, using this method of imaging myosin filaments completely abolishes the ability to correlate any observed effect with the heavy chain isoform composition of the filament.

With respect to the reviewer’s wishes to tone down the use of the word “mechanical” in the text, we respectfully would like to use it. The online Oxford dictionary includes the definition “Relating to physical forces or motion; physical”. We will endeavor to make it clear that we are only studying unloaded motility in our studies so that there is no confusion. Below are two places where we have modified the text to account for this.

“Given the strikingly different behaviors of NM2-A and NM2-B filaments described above, we next investigated how co-filaments containing both myosin paralogs moved in our assay.” Here we changed the original “behaved mechanically” to “moved”.

New text reads “Despite their almost identical structure, NM2-A and NM2-B filaments differ significantly in their motile and kinetic properties. NM2-B filaments have a higher composite duty ratio and move slower than NM2-A filaments under the unloaded conditions of the single filament in vitro motility assay. The effect of force on the kinetics of NM2 paralogs has not been measured using optical trapping,”we changed “mechanical” to “motile” and added the text “…under the unloaded conditions of the single filament motility assay.”

The only published data concerning the force sensitivity of these myosins are our stopped-flow kinetic study of ADP release which we published in Kovacs et al. (Kovacs et al., 2007). This is relevant since the release of ADP from the acto-myosin complex is likely the rate limiting step in determining attachment lifetimes. We agree with the reviewers that future work to determine the force dependence of NM2 isoforms should further elucidate the molecular mechanism of NM2 mechanics, and we plan to study this effect in the near future.

3) Please clarify in the text whether the duty ratios given assume that the two heads of one molecule interact independently with actin. “[…] the duty ratio of an NM2-B monomer" – does this mean one head?

In the first sentence under the subheading “More than four NM2-B motors are required for processive movement” we stated that: “The duty ratio of an NM2-B monomer is a function…”. This has been changed to *“*The duty ratio of a single NM2-B motor (subfragment-one)…*”* to emphasize that the determined duty ratio is for a single, independently active myosin motor.

4) The Discussion considers the potential for filaments with different isotype mixes to deal differently with strain, but again, since there are no measurements of strain here, I think the authors should restrict themselves to pointing out the possibilities, rather than implying that the current results speak directly to the role of strain versus isotype-mix. The Kovacs (2007) study was about internal strain (strain between heads of one molecule). The internal stress and strain on myosins in filaments will be different in a nontrivial way (Hariadi, Nat Nano, 2015). External strain (for example, that coming from another myosin filament pulling on the sameactin, could also be relevant in vivo.

The experiments, where mixed filaments of NM2-A and NM2-B were created, do demonstrate strain dependence since the NM2-A molecule crossbridge kinetics will be modified by slower NM2-B crossbridge kinetics and vice versa. This mixed myosin assay is no longer “unloaded”. Similar to experiments carried out with mixed molecules in the gliding filament assay, the slower myosin tends to dominate the velocity (Cuda et al., 1997). We have modified a sentence at the end of the Results to emphasize this: “This was consistent with previous results from actin gliding in vitro motility assays, which show that slower myosins dominate the velocity when myosin 2 isoforms of different inherent velocity are randomly bound to the coverslip surface which was modeled by assuming the two myosin isoforms affect the attachment lifetimes of each other (Cuda et al., 1997).*”*

In many aspects the Hariadi et al. paper on synthetic “filaments” agrees with our work. Although most of the work in Hariadi et al. was done with the processive class V and VI myosins, the study showed that the velocity of actin filament gliding was not dependent on the number of crossbridges participating nor on the spacing of these crossbridges. The fact that reduction in the number of possible myosin motors by copolymerizing with a headless tail fragment does not diminish speed of movement is consistent with Hariadi et al. as well as many studies from the past. Hariadi et al. also studied the effect of changing the compliance which we are unable to do since we are dealing with a native thick filament. We have included a reference to this work: “The observation that the rate of movement of the filaments did not vary with the number of contributing motors was consistent with a recent study which made synthetic myosin filaments mimics using DNA nanotubes where the spacing and density of myosin motor domains could be varied. (Hariadi et al., 2015).”

The effect of strain on myosin filament kinetics is of great interest to us. The Kovacs et al. (Kovacs et al., 2007) study used the fact that the two motors of an NM2 HMM molecule can simultaneously bind to two adjacent actin monomers in the presence of ADP to look at the effect of strain on ADP release, which is likely to be the kinetic step that controls cycling rate. Note that another study from our lab (Nagy et al., 2013) demonstrated that in the presence of ATP, it is unlikely that NM2-B binds to actin via both motors at the same time, so this intramolecular strain between two bound motors from the same myosin is not likely to be a source of internal strain in our filament experiments. However, motors from different myosins within a filament could very likely produce internal strain. We have no mechanisms in the present study to determine the number of myosin motors within a filament that simultaneously interact with actin. This is a point of great interest to us for future studies.

It is not likely that external strain (coming from another myosin filament pulling on the same actin filament) is a factor in most of the experiments presented in the current manuscript since the actin filaments are typically bound to the coverslip surface and are not free to move. The reviewer is correct however, in that external strain caused by another myosin filament acting on the same actin filament could be another factor in cells. Strain could also be generated if an actin filament is in contact with actin binding proteins that are anchored to the cytoskeletal or the membrane. We have included a sentence in the Discussion to emphasize this:

“Other sources of strain in a cell might affect the mechanical properties of NM2 paralogs. Strain could arise from two myosin filaments interacting with the same actin filament or from actin binding proteins that might tether the filament to the cytoskeletal component or a membrane which might provide resistance to the NM2 directed motion.”

5) Please comment, if possible, on whether exchange of fluorescent molecules between filaments occurs on a relevant timescale.

This is a good point and is something that we plan to be study this in detail. In each mixing experiments, both filament components were covalently labeled with different color HaloTags. This allows us to look at the ratio of the two colors of filaments that are moving processively along actin. We monitored the intensity of the two colors of single myosin filaments during their attachment time to actin, which can be as much as 10 min, and typically only observed gradual photobleaching of each color. Since the photobleaching rates of the two dyes were not identical, this did result in a slight change in the ratio of the two colors, but we saw no evidence where one paralog was being substituted for the other during the course of any given processive run for a single filament. This was not a great surprise to us since the free myosin concentration in our experiments is quite low. Whether there is exchange occurring in solution between myosin filaments that are unbound to actin is not relevant for our experimental set up since we are determining the myosin composition of each moving filament while quantifying its velocity and run lengths. The traces are noisy. Some of the contribution to this might be that the filaments change their orientation on actin while moving and are long compared to the depth of the TIRF field. This could result in fluctuations in intensity. We observe the occasional trace where the intensity of one color rapidly spikes, but this is probably caused by some artifact, such as two filaments approaching within the diffraction limited distance, since the spikes are large and rapid and would represent a major change in the filament composition occurring in a very short interval of time. We include this data below for the reviewers’ benefit (see Author response image 1). We included a sentence to this effect in the second paragraph: “We did not see evidence for the filaments changing their isoform composition during an actin-attached processive run.”

**Author response image 1. respfig1:** Each color shows the proportion of NM2-A in a given filament during a processive run.

6) A paper by Diensthuber et al., (2011) suggests that phalloidin stabilization of actin inhibits the motile properties of NM2-A, but not NM2-B. Phalloidin is used in these experiments, so the authors should address this point about possible inhibition of the motor.

In response to the reviewers’ request, we have performed additional experiments to study the ability of NM2-A HMM to translocate Atto-565-labeled actin, and also actin labeled with two different phalloidin derivatives, rhodamine and Alexa-Fluor-647. There was no significant difference in the rate of actin filament gliding with any of these actins. We include this data as Supplementary file 3 and have added a sentence: “Note, that in contrast to a previous study (Diensthuber et al., 2011), we found no effect of phalloidin on the movement of actin filaments by NM2A (Supplementary file 3).”

7) The Materials and methods do not reveal how the authors deal with limitation of actin filament length in calculating the run length.

In the original manuscript under the heading of “data analysis” we described this as follows: *“*NM2 filaments that reached the end of the actin filament and dissociated were excluded from the analysis.*”*

We now phrase it somewhat differently under the same heading: “Only myosin filaments that were observed to bind to actin, move and dissociate before reaching the end of the actin filament were included in the analysis. Thus, we are measuring a minimum estimate of the run length”.

8) The meaning of uncertainties are not always clear in the figures. For example, Figure 1 shows a distribution with a mean +/- 1 nm. I assume this is S.E.M. I would actually prefer to see the STD of the distributions throughout the manuscript, since that is the more useful parameter when comparing the different distributions. For example, the difference in the widths of the distributions between Figure 1 and Figure 3 is interesting.

We have now endeavored to make sure that the definition of the errors is given in each legend. We have given the standard deviation and the standard error in each figure legend for the velocity measurements, but have left the error bars for the velocity determinations in the plots as S.E.M.s. In reviewing the single molecule motility as well as the gliding actin in vitro motility literature, we have observed that most investigators present the run length error as S.E.M. and there is an increasing tendency to plot the velocity data in both types of assays with S.E.M.s as the error bars. We have also added R^2^ values to the figure legends.

9) Figure 2, third panel: It is difficult to believe that there can be confidence in the exponential fit of these binned data. Is this binning actually what was used to obtain the fit? Fitting to smaller bins, cumulative distributions, or via MLE would be more appropriate.

We investigated the effect of bin size on the exponential fit to the data and found that changing the bin size to 0.5nm for the right hand plot in Figure 2 distributed the data points better and gave a reasonable fit. The values were not greatly changed. We also plotted cumulative frequency distributions, but were not satisfied with the fits. The change in bin size for the third panel in Figure 2 did not dramatically change the value for run length, but we modified Figure 2 to reflect this and received a slightly different “minimum number of heads” of 4.1 vs. 4.9. The text has been modified to reflect this.

10) The rates of filament motility appear to be limited by the rate at which myosins detach from actin (Figure 2), which is at odds with the results of Brizendine et al., (2015), which suggest an attachment limited model. It would be helpful to the myosins specialists if this point was discussed.

The Brizendine et al. (Brizendine et al., 2015) study differs from ours in several important ways. First, they were using smooth muscle myosin which makes side polar filaments and has a lower duty ratio. Second, since these filaments did not move processively on actin in the absence of methycellulose, they used this viscosity enhancing reagent in all their assays. In the experiments relative to ours (where we mixed myosin tail fragments and full length NM2-B molecules to reduce myosin motor density) we did not use methylcellulose since this reagent is not required for the motility of this myosin. As noted in our manuscript NM2-A also does not move processively in the absence of methylcellulose, but does so in its presence. We believe this is because methylcellulose effectively limits the diffusion of myosin away from the actin filament such that if the filament is transiently detached it will not diffuse sufficiently far away that another myosin motor cannot rebind to keep the myosin filament tethered to the actin filament. Thus, the NM2-A filament (or the smooth muscle myosin filament) in the presence of methylcellulose may not truly be moving “processively” through its entire “observed” run length as we and others have defined processivity for single molecule studies using two-headed myosin5 or myosin-6 constructs. This definition requires that the myosin take multiple steps on actin per diffusional encounter and assumes that the molecule diffuses away when neither of its two motors is attached to actin.

A second paper recently published from this lab (Brizendine et al. 2017) elaborated on their model and suggested that whether a filament’s motility was limited by myosin attachment or detachment would vary with the conditions and possible source of myosin. We have now referenced this paper and modified the text accordingly. We believe that determination of the limiting step for movement in motility assays and muscles is still not firmly established and merits further work. We have added new discussion on this point to the revised manuscript:

“We also found that the rate of movement of the hybrid myosin filaments on actin were not significantly affected by the number of myosin motors available. […] The observation that the rate of movement of the filaments did not vary with the number of contributing motors was consistent with a recent study which made synthetic myosin filaments mimics using DNA nanotubes where the spacing and density of myosin motor domains could be varied. (Hariadi et al., 2015).”

11) The finding that myosin filaments can form stacks in vitro is interesting. Can the authors speculate on the mechanism by which the stacks form?

When unphosphorylated NM-2 forms the folded monomeric inactive conformation the two heads of the molecule make an asymmetric contact wherein the loop 2 region of one motor contacts the converter region of the other (Wendt et al., 2001) (Jung et al., 2008). This contact appears to be evolutionarily conserved across many phyla. Our speculation is that this “weak affinity” of loop 2 for the converter domain persists when the two heads become separated upon phosphorylation and that when one filament comes in close proximity to another, potentially bringing sixty myosin heads in close proximity that the cumulative attraction of these multiple weak interactions are sufficient for aligning the filaments in a stack. We have elaborated on this in the revised Discussion:

“The physical basis for forming stacks is not known. […] The fact that the stacks can disassemble suggests that the forces holding them together are not large.”

12) Subsection “More than five NM2B motors are required for processive movement”: The authors' discussion of the duty ratio does not consider head-head gating. Although this is mentioned later in the manuscript, it is appropriate to mention that this equation does not take this process into account.

This is indeed true. Please note however, as discussed in response to question 4, our previous data strongly supports the fact that NM2-B molecules are unlikely to bind to actin via two motors of the same molecule in the presence of ATP (Nagy et al., 2013) and so the type of gating that was presented in Kovacs et al. (Kovacs et al., 2007) is unlikely to occur in these filaments. We have included a couple of sentences to explain this:

“This equation assumes that each motor is capable of interacting with actin. In the presence of ATP, NM2-B molecules only interact with actin via one motor at a time (Nagy et al., 2012) so corrections for intrahead gating as shown by Kovacs et al. (Kovacs et al., 2007) do not have to be considered.”

13) Discussion, last two paragraphs: The wording in the paragraphs is awkward. The topic sentence of the first paragraph is: "Our results raise experimental concerns in elucidating the role of NM2s in cell biological experiments." However, it is not clear from the paragraph which concerns were raised by the results. The information in the second paragraph actually seems to be more related to this sentence.

We have rewritten this passage and split it into two paragraphs to hopefully make it clearer:

“Our results suggest that relating light microscopic observations of myosin filaments in cells to their mechanical output is not trivial. […] Both labeling techniques are currently used in live cell imaging experiments (Beach et al., 2017; Bruun et al., 2017).*”*